# On the Stability of Graph Convolutional Neural Networks: A Probabilistic Perspective

**Ning Zhang**
University of Oxford
ning.zhang@stats.ox.ac.uk

**Henry Kenlay**
Independent Researcher*
henrykenlay@pm.me

**Li Zhang**
University College London
ucesl07@ucl.ac.uk

**Mihai Cucuringu**
UCLA, University of Oxford
mihai@math.ucla.edu

**Xiaowen Dong**
University of Oxford
xdong@robots.ox.ac.uk

## Abstract

Graph convolutional neural networks (GCNNs) have emerged as powerful tools for analyzing graph-structured data, achieving remarkable success across diverse applications. However, the theoretical understanding of the stability of these models, i.e., their sensitivity to small changes in the graph structure, remains in rather limited settings, hampering the development and deployment of robust and trustworthy models in practice. To fill this gap, we study how perturbations in the graph topology affect GCNN outputs and propose a novel formulation for analyzing model stability. Unlike prior studies that focus only on worst-case perturbations, our distribution-aware formulation characterizes output perturbations across a broad range of input data. This way, our framework enables, for the first time, a probabilistic perspective on the interplay between the statistical properties of the node data and perturbations in the graph topology. We conduct extensive experiments to validate our theoretical findings and demonstrate their benefits over existing baselines, in terms of both representation stability and adversarial attacks on downstream tasks. Our results demonstrate the practical significance of the proposed formulation and highlight the importance of incorporating data distribution into stability analysis.

## 1 Introduction

The past decade has witnessed an explosion of interest in analyzing data that resides on the vertices of a graph, known as *graph signals* or *node features*. Graph signals extend the concept of traditional data defined over regular Euclidean domains to the irregular structure of graphs. To facilitate machine learning over graph signals, classical convolution neural networks have been adapted to operate on graph domains, giving rise to a class of graph-based machine learning models – graph convolutional neural networks (GCNNs) [9, 27, 52, 28, 13]. At the heart of GCNNs lies the use of *graph filters*, which aggregate information from neighboring vertices in a way that respects the underlying graph structure [22, 12]. This mechanism enables GCNNs to produce structure-aware embeddings that effectively integrate information from both the input signals and the graph structure.

As GCNNs become prevalent in real-world applications, understanding their robustness under graph perturbations has become increasingly important. In particular, to ensure the deployment of trustworthy models, it is essential to assess their *inference-time stability*, how sensitive a pre-trained GCNN's predictions are to a small input perturbation. This is especially critical because real-world networks could differ from the clear and idealized samples seen during training. For instance, in social

---

*Work done while at the University of Oxford.

39th Conference on Neural Information Processing Systems (NeurIPS 2025).

networks, adversarial agents such as bot accounts can strategically modify the network structure, by adding or removing connections, to mislead a pre-trained GCNN-based fake news detector [48]. This consideration motivates our investigation into how structural perturbations affect the predictions of pre-trained GCNNs.

In this paper, we focus specifically on analyzing the embedding stability – the sensitivity of GCNN output embeddings to perturbations in the graph structure. Analyzing stability at the embedding level provides a task-agnostic perspective that avoids assumptions about specific downstream objectives (e.g., classification or regression), making our analysis broadly applicable across different model architectures and application domains. A line of recent studies on the embedding stability of graph filters and GCNNs has primarily adopted a worst-case formulation, analyzing the maximum possible change in embeddings under edge perturbations [29, 30, 23, 25, 35]. However, a significant limitation of this worst-case view is its incompleteness: it overlooks output embeddings that are produced by graph inputs other than extreme cases; therefore, it could be overly pessimistic, as seen in Figure 1. This motivates us to explore alternative formulations that capture a wide spectrum of embedding perturbations beyond the worst-case scenarios, rendering the analysis amenable to more realistic data. We summarize our main contributions as follows:

*A probabilistic formulation (Section 3).* We propose a probabilistic framework for analyzing the embedding stability of graph filters and GCNNs under edge perturbation. Unlike prior work that focuses on the worst-case scenarios, we introduce the notion of expected embedding perturbation, enabling a more representative assessment of stability across a broad range of input graph signals. Under this framework, we derive an exact characterization of the stability of graph filters and an upper bound for multilayer GCNNs. Both results are distribution-aware, revealing how embedding deviations are jointly determined by the second-moment matrix of graph signals and graph topology perturbations. To the best of our knowledge, this is the first work to establish a framework for understanding the interplay between data distribution and structural perturbations in the context of embedding stability.

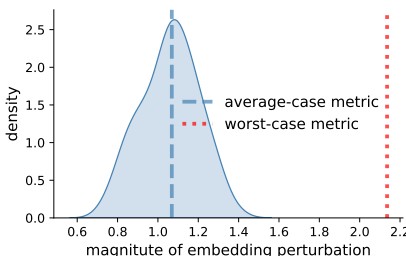

Figure 1: Embedding perturbations of a graph filter on Zachary's karate club network with 100 randomly sampled unit-length graph signals.

*A structural interpretation (Section 4).* We further interpret our theoretical results to better understand how the interaction between graph perturbation and signal correlation influences model stability. Such interpretation offers insight into why perturbations to certain parts of the graph structure are more impactful than others, as empirically observed in recent studies [59, 46]. Through a case study on the contextual stochastic block model (cSBM), we show that our framework offers a rigorous explanation for heuristic perturbation strategies, clarifying, through the lens of embedding perturbation, why and when they are effective.

*Application in adversarial attacks (Section 5).* Based on our theoretical results, we propose `Prob-PGD`, an efficient projected gradient descent method for identifying highly impactful edge perturbations. Through extensive experiments, we demonstrate that `Prob-PGD` consistently produces on average larger embedding perturbations compared to existing baselines. Despite the focus on embedding stability, we further demonstrate that our method leads to greater performance degradation in downstream tasks, including node and graph classification using GCNNs.[2]

**Related work.** Analyzing the inference-time stability of GCNNs has gained growing interest in recent years due to its popularity in a variety of real-world applications. Existing theoretical studies investigate different types of perturbations: Gao et al. [18], Testa et al. [43, 44], Wang et al. [49], Ceci and Barbarossa [7] study the embedding perturbation induced by stochastically adding or deleting a small number of edges. In Liao et al. [32], the authors analyzed the change in embeddings of GCNNs under a perturbation applied to the learnable model parameters. Ruiz et al. [39] adopted a relative perturbation model where the perturbation on the graph is formulated as a multiplicative factor applied to the graph shift operator. Keriven et al. [26] studies the stability of GCNNs to small deformations of the random graph model. Our work is most closely related to a series of studies on the embedding

---

[2]Code is available at: `https://github.com/NingZhang-Git/Stability_Prob`

stability under deterministic edge perturbation on the graph [29, 30, 23, 25, 16, 35, 37, 38]. The main difference between our work and existing embedding stability studies is that prior analyses are limited to the worst-case embedding perturbations, while our work first explores a more representative notion using expected embedding perturbations. Such formulation not only captures embedding behaviour from a broader range of inputs, but also enables the integration of input distribution into the stability analysis.

Finally, we note that stability analysis in graph machine learning offers another perspective on robustness, known as *training stability*, which examines how perturbations to the training dataset influence the learning dynamics [45, 59, 4, 8]. While conceptually related, training stability concerns a different aspect of robustness and is less directly connected to our focus on the post-training behavior of models under input perturbations.

## 2 Preliminaries and problem formulation

### 2.1 Notation and definitions

**Graphs and Graph Signals.** Let $\mathcal{G} = (\mathcal{V}, \mathcal{E}, \mathbf{A})$ be a graph, where $\mathcal{V}$ is the set of vertices, $\mathcal{E}$ is the set of edges, and $\mathbf{A} \in \mathbb{R}^{n \times n}$ is the (possibly weighted) adjacency matrix. We denote $\mathcal{G}_p = (\mathcal{V}, \mathcal{E}_p, \mathbf{A}_p)$ a perturbed version of $\mathcal{G}$, where only the edge set is modified, while the vertex set remains unchanged. A graph signal refers to a function mapping from the vertex set $\mathcal{V}$ to a real value, which can be equivalently defined as a vector $\mathbf{x} \in \mathbb{R}^n$. When multiple graph signals are considered, we represent them by a matrix $\mathbf{X} \in \mathbb{R}^{n \times d}$, where each column $\mathbf{X}_{:,i}$ corresponds to an individual signal.

**Graph Filters.** The structural information of a graph $\mathcal{G}$ is encoded via a graph shift operator $\mathbf{S}$, a self-adjoint matrix satisfying $\mathbf{S}_{i,j} = 0$ for all $(i, j) \notin \mathcal{E}$. Common choices for $\mathbf{S}$ include the graph adjacency matrix $\mathbf{A}$ and the graph Laplacian $\mathbf{L}$ [5]. A graph filter is defined as a function of the graph shift operator, denoted as $g(\mathbf{S})$. Typical examples of graph filters include polynomial filters, $g(\mathbf{S}) = \sum_{k=0}^{K} c_k \mathbf{S}^k$ and autoregressive moving average filter [21]. Given input graph signals $\mathbf{X}$, the filter output (i.e., output embedding) is expressed as $g(\mathbf{S})\mathbf{X}$.

This paper investigates the impact of edge perturbations on the outputs of graph filters. Specifically, we measure the change in the output embedding using the squared Frobenius norm $\|g(\mathbf{S})\mathbf{X} - g(\mathbf{S}_p)\mathbf{X}\|_F^2$, where $\mathbf{S}_p$ denotes the graph shift operator of the perturbed graph $\mathcal{G}_p$. Throughout our analysis, we fix the filter function $g(\cdot)$ and consider changes in the filter output only as a consequence of the edge perturbation on the graph. For notational simplicity, when the specific forms of $\mathbf{S}$ and filter function are not essential to the context, we use the shorthand $\mathbf{E}_g = g(\mathbf{S}) - g(\mathbf{S_p})$ to denote the filter difference. Accordingly, the embedding perturbation of input $\mathbf{X}$ is denoted as $\|\mathbf{E}_g \mathbf{X}\|_F^2$. Concrete examples illustrating these concepts are provided in Appendix A.2.

**Graph Convolutional Neural Networks.** GCNNs are machine learning models that extend convolutional neural networks to graph-structured data by leveraging graph filters [17]. They consist of cascaded layers of graph filters $g(\mathbf{S})$ followed by a non-linear activation function $\sigma(\cdot)$. Given a graph $\mathcal{G}$ and an input feature matrix $\mathbf{X}^{(0)}$, the embeddings at each layer are computed recursively as

$$\mathbf{X}^{(l)} = \sigma^{(l)} \left( g(\mathbf{S}) \mathbf{X}^{(l-1)} \mathbf{\Theta}^{(l)} \right), \tag{1}$$

where $\sigma^{(l)}(\cdot)$ denotes the activation function at layer $l$, and $\mathbf{\Theta}^{(l)} \in \mathbb{R}^{d_{l-1} \times d_l}$ is the learned parameter in the $l$-th layer. Throughout our analysis, we assume the model parameters $\{\mathbf{\Theta}^{(l)}\}_{l=1}^{L}$ are fixed after training and do not change at test time. Therefore, any change in the final embedding is attributed solely to perturbations in the underlying graph $\mathcal{G}$. The embedding perturbation for an $L$-layer GCNN is measured by $\|\mathbf{X}^{(L)} - \mathbf{X}_p^{(L)}\|_F^2$, where $\mathbf{X}^{(L)}$ and $\mathbf{X}_p^{(L)}$ are computed via the recursion in (1) using graph filters $g(\mathbf{S})$ and $g(\mathbf{S}_p)$, respectively. We summarize common GCNN architectures and their associated graph filter functions in Table 4 in Appendix A.2.

### 2.2 Problem formulation

In this paper, we study the stability of graph filters and GCNNs from a probabilistic perspective. Specifically, we consider random input $X \in \mathbb{R}^{n \times d}$, where each column $X_{:,i}$ corresponds to a graph signal drawn independently and identically from an unknown distribution $\mathcal{D}$ over $\mathbb{R}^n$. We make no assumptions about the form of $\mathcal{D}$ beyond its second-order statistics, which are captured by the second-moment matrix $\mathbf{K} = \mathbb{E}[X_{:,i} X_{:,i}^T]$ (also referred to covariance matrix when $\mathbb{E}[X_{:,i}] = \mathbf{0}$).

Since the columns $X_{:,i}$ are i.i.d. samples from $\mathcal{D}$, this second-moment matrix is the same for all $i$. To evaluate output stability, we introduce the notion of expected embedding perturbation.

**Definition 1.** *Let $g(\mathbf{S})$ be a graph filter. Its embedding stability with respect to a signal distribution $\mathcal{D}$ is measured by the expected embedding perturbation given by*

$$\mathbb{E}_{X \sim \mathcal{D}}[\|g(\mathbf{S})X - g(\mathbf{S}_p)X\|_F^2] = \mathbb{E}_{X \sim \mathcal{D}}[\|\mathbf{E}_g X\|_F^2]. \tag{2}$$

*For an $L$-layer GCNN defined as in* (1) *with graph filter $g(\mathbf{S})$, its embedding stability with respect to signal distribution $\mathcal{D}$ is quantified by*

$$\mathbb{E}_{X \sim \mathcal{D}}[\|X^{(L)} - X_p^{(L)}\|_F^2]. \tag{3}$$

Under this probabilistic framework, our goal is to characterize the expected embedding perturbations of graph filters (2) and GCNNs (3), and to understand how they depend on the signal distribution $\mathcal{D}$ as well as the underlying graph. As our discussion involves both random and deterministic graph signals, we adopt the convention that boldface letters represent deterministic vectors or matrices (e.g., $\mathbf{x}, \mathbf{X}$), while uppercase letters without boldface denote random vectors or matrices (e.g., $X$). We provide a summary of notations in Appendix A.1 for ease of reference.

**Remark 1.** *Our probabilistic formulation is general as it does not impose any assumptions on the generative distribution of graph signals. Particularly, by specializing the signal distribution to let the random graph signal $X$ assign all probability mass to the worst-case scenarios, our analysis encompasses the existing worst-case analysis, such as Kenlay et al. [25], as a special case.*

**Remark 2.** *To maintain focus, we do not explicitly address the permutation equivariance of graph filters and GCNNs, which is considered in other stability studies such as Gama et al. [16]. However, our framework can readily incorporate such equivariance by only replacing the filter perturbation $\mathbf{E}_g = g(\mathbf{S}) - g(\mathbf{S}_p)$ with its permutation-modulated counterpart $\mathbf{E}_g = g(\mathbf{S}) - g(\mathbf{P}_0 \mathbf{S}_p \mathbf{P}_0^T)$ where $\mathbf{P}_0 = \arg\min_{\mathbf{P} \in \Pi_n} \|\mathbf{P}^T \mathbf{S} \mathbf{P} - \mathbf{S}_p\|_F$.*

# 3 Analyzing Embedding Stability of Graph Filters and GCNNs

Under our probabilistic formulation, this section studies the stability of graph filters and GCNNs by characterizing their expected embedding change under edge perturbations. We begin in Section 3.1 with the analysis of graph filters, which are simpler linear models that serve as foundational components of GCNNs. Building on this, Section 3.2 extends the analysis to the more complex case of multilayer GCNNs.

## 3.1 Stability Analysis of Graph Filters and Single-Layer GCNNs

**Theorem 1** (Stability of Graph Filters). *Let $X \in \mathbb{R}^n$ be a random graph signal from a distribution $\mathcal{D}$ with second moment matrix $\mathbf{K}$. Then for any graph filter perturbation $\mathbf{E}_g = g(\mathbf{S}) - g(\mathbf{S}_p)$, the expected change in output embedding is*

$$\mathbb{E}_{X \sim \mathcal{D}}[\|\mathbf{E}_g X\|_F^2] = \langle \mathbf{K}, \mathbf{E}_g^T \mathbf{E}_g \rangle. \tag{4}$$

**Corollary 1.** *For any $c > 0$, we have $\mathbb{P}\left(\|\mathbf{E}_g X\|_F^2 \geq (1 + c)\langle \mathbf{K}, \mathbf{E}_g^T \mathbf{E}_g \rangle\right) \leq \frac{1}{1+c}$.*

In Theorem 1, we provide an exact characterization of the expected embedding perturbation in (4). Furthermore, Corollary 1 establishes a concentration bound that quantifies the probability of deviation from the expected value, thereby reinforcing the utility of our notion of stability. The proof of Theorem 1 is provided in Appendix B.1. Detailed comparisons between the expected-case and worst-case embedding perturbation metrics are presented in Appendix C.4.

Equation 4 shows that the expected embedding perturbation of a graph filter is equivalent to the filter perturbation $\mathbf{E}_g$ measured in a $\mathbf{K}$-induced norm, i.e., $\langle \mathbf{K}, \mathbf{E}_g^T \mathbf{E}_g \rangle = \text{Tr}(\mathbf{E}_g \mathbf{K} \mathbf{E}_g^T)$. This expression re-weights the perturbation according to the second-moment matrix, highlighting the interplay between graph filter perturbation and the correlation structure of the input graph signals. This perspective becomes particularly interesting when the second-order statistics of graph signals are correlated with the graph topology, which is a commonly adopted assumption in the field of graph signal processing [57, 34] and machine learning [2]. A more detailed discussion of how this correlation influences stability is deferred to Section 4.

Recall that for a single-layer GCNN, the embedding of multiple signals $X \in \mathbb{R}^{n \times d}$ is given by

$$X^{(1)} = \sigma \left( g(\mathbf{S}) X \mathbf{\Theta}^{(1)} \right), \quad X_p^{(1)} = \sigma \left( g(\mathbf{S}_p) X \mathbf{\Theta}^{(1)} \right)$$

where $\sigma(\cdot)$ is a pointwise activation function and $\mathbf{\Theta}^{(1)}$ is a learned weight matrix. Due to the non-linearity of the activation function, we can no longer derive an exact expression for the expected embedding perturbation. Instead, Corollary 2 provides a distribution-dependent upper bound. The proof is deferred to Appendix B.3.

**Corollary 2** (Stability of Single-Layer GCNNs). *Let $X \in \mathbb{R}^{n \times d}$ be a random matrix whose columns are i.i.d. graph signals from a distribution $\mathcal{D}$ over $\mathbb{R}^n$, with second-moment matrix $\mathbf{K}$, and $\sigma(\cdot)$ be $C_\sigma$-Lipschitz continuous. Then the expected embedding perturbation of a single-layer GCNN satisfies*

$$\mathbb{E}_{X \sim \mathcal{D}}[\|X^{(1)} - X_p^{(1)}\|_F^2] \leq d C_\sigma^2 \|\mathbf{\Theta}^{(1)}\|^2 \langle \mathbf{K}, \mathbf{E}_g^T \mathbf{E}_g \rangle.$$

We next establish a connection between the embedding perturbation of a single-layer GCNN and that of its associated graph filter. The following result holds under a mild monotonicity condition on the activation function.

**Corollary 3.** *Let $\sigma(\cdot)$ be a monotonically non-decreasing activation function. Then the expected embedding perturbation of a single-layer GCNN is monotonically non-decreasing in the expected embedding perturbation of its associated graph filter.*

Corollary 3 implies that for a single-layer GCNN with monotonically non-decreasing activation functions, any optimization task involving the embedding perturbations can be reduced to an equivalent problem over the associated graph filter. This observation is particularly valuable for studying adversarial edge perturbations, which are commonly formulated as optimization tasks. Since most commonly used activation functions, such as ReLU and Sigmoid, are monotonically non-decreasing, this corollary simplifies the analysis of adversarial attacks by allowing us to focus only on the embedding perturbation of graph filters. For the later case, which is linear, Theorem 1 provides an exact expression for the expected perturbation. As a result, this connection enables near-exact stability analysis for a broad class of single-layer GCNNs, including models such as SGC [52], SIGN [14], and gfNN [36].

### 3.2 Stability Analysis of Multilayer GCNNs

We now extend our analysis to multilayer GCNNs, which generalize the single-layer case by stacking multiple layers of graph filtering $g(\mathbf{S})$, nonlinear activation $\sigma(\cdot)$, and learned model parameters $\{\mathbf{\Theta}^{(j)}\}_{j=1}^L$. Unlike the single-layer setting, multilayer architectures introduce recursive dependencies and compound nonlinearities, making exact characterization of the embedding perturbation analytically intractable. Nevertheless, by leveraging the results for graph filters and carefully bounding the propagation of perturbations across layers, we derive in Theorem 2 an upper bound on the expected embedding change for $L$-layer GCNNs.

Throughout our analysis of multilayer GCNNs, we make the following standard assumptions:

$$\|g(\mathbf{S})\| \leq C \quad \text{and} \quad \|g(\mathbf{S}_p)\| \leq C \tag{A1}$$

$$\sigma^l(\cdot) \text{ is } C_\sigma\text{-Lipschiz continuous for all } 1 \leq l \leq L. \tag{A2}$$

$$\sigma^l(0) = 0 \text{ for all hidden layer } 1 \leq l \leq L-1. \tag{A3}$$

Assumption (A1) requires that the spectral norm (largest singular value) of $g(\mathbf{S})$ and $g(\mathbf{S}_p)$ are bounded by a constant $C$. This condition is trivially satisfied for finite graphs and can also be ensured for infinite graphs through appropriate normalization of the filter, such as dividing by the maximum degree. Assumption (A2) imposes Lipschitz continuity on the activation functions, a property satisfied by most commonly used functions such as ReLU ($C_\sigma = 1$), Tanh ($C_\sigma = 1$), and Sigmoid ($C_\sigma = 1/4$). Assumption (A3) requires that hidden-layer activation functions output zero when their input is zero, which is satisfied by many standard hidden-layer activation functions, including ReLU, Leaky ReLU, and Tanh.

**Theorem 2** (Stability of $L$-layer GCNN). *Let $X \in \mathbb{R}^{n \times d}$ be a random matrix whose columns are i.i.d. graph signals from a distribution $\mathcal{D}$ over $\mathbb{R}^n$, with second-moment matrix $\mathbf{K}$. Consider*

an $L$-layer GCNN built on a graph filter $g(\mathbf{S})$ and let the corresponding filter perturbation be $\mathbf{E}_g = g(\mathbf{S}) - g(\mathbf{S}p)$. Suppose the model satisfies assumptions (A1) (A2) and (A3). Then the expected embedding perturbation of the $L$-layer GCNN satisfies

$$\mathbb{E}_{X \sim \mathcal{D}}\Big[\|\mathbf{X}^{(L)} - \mathbf{X}_p^{(L)}\|_F^2\Big] \leq d\, \underbrace{C_\sigma^{2L} C^{2L-2} \prod_{j=1}^{L} \|\mathbf{\Theta}^{(j)}\|^2}_{model} \Big(\underbrace{(L-1)}_{model}\underbrace{\|\mathbf{E}_g\|^2}_{pert.}\underbrace{\text{tr}(\mathbf{K})}_{signals} + \underbrace{\langle \mathbf{K}, \mathbf{E}_g^T \mathbf{E}_g \rangle}_{K\text{-modulated pert.}}\Big). \quad (5)$$

Our result in (5) shows that the expected embedding perturbation of an $L$-layer GCNN is governed by several key components. The first is model complexity, captured by the factor $C_\sigma^{2L} C^{2L-2} \prod_{j=1}^{L} \|\mathbf{\Theta}^{(j)}\|^2$ and $(L-1)$, which reflects the cumulative effects of activation non-linearity, filter norms, and learned weights. The bound also depends on the filter perturbation norm $\|\mathbf{E}_g\|^2$ and trace of the second-moment matrix $\text{Tr}(\mathbf{K})$, which reflect the magnitudes of the filter change and input intensity, respectively. The second-moment matrix modulated filter perturbation term $\langle \mathbf{K}, \mathbf{E}_g^T \mathbf{E}_g \rangle$, is derived from bounding the recursive propagation of perturbations across layers. This term highlights how the embedding change is shaped jointly by the graph filter perturbation and the second-order statistics of the graph signal. For further intuition and a complete derivation, we refer the reader to the proof in Appendix B.4.

In prior work, Kenlay et al. [24] showed that for unit-length input signals, the worst-case embedding perturbation of an $L$-layer GCN [27] satisfies $\sup \|\mathbf{x}^{(L)} - \mathbf{x}_p^{(L)}\|_F^2 \leq dL^2 \prod_{j=1}^{L} \|\mathbf{\Theta}^{(j)}\|^2 \|\mathbf{E}_g\|^2$. Compared to this worst-case bound, our result presents a tighter characterization of the embedding stability. More importantly, our result is distribution-aware, capturing the joint influence of second-order statistics and graph topology perturbations on the embedding deviation.

## 4 A Structural Interpretation

Our theoretical results demonstrate that the expected embedding perturbations of both graph filters and GCNNs are crucially related to the second-moment matrix modulated filter perturbation term $\langle \mathbf{K}, \mathbf{E}_g^T \mathbf{E}_g \rangle$, which is jointly determined by the signal correlation and the structure of the filter perturbation. In this section, we take one step further toward a structural interpretation, understanding which parts of the graph, when perturbed, have the greatest impact on embeddings. In particular, we aim to understand how the interplay between graph structure and signal correlation influences model stability. This perspective offers valuable insight into understanding why certain perturbations are more impactful than others, as empirically observed in several recent studies [31, 46, 50].

The filter perturbation $\mathbf{E}_g = g(\mathbf{S}) - g(\mathbf{S}_p)$ depends on the specific choice of graph filters, therefore, a unified analysis is not feasible, and an exhaustive enumeration of all possible filters is impractical. We therefore focus on two representative graph filters: a low-pass filter using graph adjacency matrix $g(\mathbf{S}) = \mathbf{A} \in \{0,1\}^{n \times n}$, and a high-pass filter using the graph Laplacian matrix $g(\mathbf{S}) = \mathbf{L} = \mathbf{D} - \mathbf{A}$. We denote $\mathcal{P} = \{\{u, v\} \subset \mathcal{V} : \{u, v\} \text{ is perturbed}\}$ the set of perturbed vertex pairs. We define $\sigma_{uv}$ to indicate the type of perturbation between vertex $u$ and $v$: $\sigma_{uv} = 1$ corresponds to adding an non-existing edge between $u$ and $v$, while $\sigma_{uv} = -1$ corresponds to deleting an existing edge.

**Proposition 1.** *Consider an edge perturbation set $\mathcal{P}$. If the graph filter is the adjacency matrix, i.e., $g(\mathbf{S}) = \mathbf{A}$, then the expected embedding perturbation satisfies*

$$\mathbb{E}_{X \sim \mathcal{D}}[\|\mathbf{E}_g X\|_2^2] = \langle \mathbf{K}, \mathbf{E}_g^T \mathbf{E}_g \rangle = \sum_{\{u,v\} \in \mathcal{P}} (\mathbf{K}_{uu} + \mathbf{K}_{vv}) + 2 \sum_{\{u,v\},\{u,v'\} \in \mathcal{P}} \sigma_{uv}\sigma_{uv'}\mathbf{K}_{vv'}. \quad (6)$$

*If the graph filter is the graph Laplacian, i.e., $g(\mathbf{S}) = \mathbf{L}$*

$$\mathbb{E}_{X \sim \mathcal{D}}[\|\mathbf{E}_g X\|_2^2] = \langle \mathbf{K}, \mathbf{E}_g^T \mathbf{E}_g \rangle = 2\sum_{\{u,v\} \in \mathcal{P}}\mathcal{R}(u,v) + \sum_{\{u,v\},\{u,v'\} \in \mathcal{P}} \sigma_{uv}\sigma_{uv'}(\mathcal{R}(u,v) + \mathcal{R}(u,v') - \mathcal{R}(v,v')),$$

$$(7)$$

*where $\mathcal{R}(u,v)$ is defined as $\mathcal{R}(u,v) \triangleq \mathbb{E}[(X_u - X_v)^2]$.*

Proposition 1 shows that the expected embedding perturbation decomposes into two parts: self-terms, which capture the individual contribution of each perturbed edge (the first summation in (6) and (7)),

and coupling terms, which capture interactions between intersecting edge pairs, i.e., edge pairs that share a common vertex (the second summation in each expression). While each self-term is non-negative, the coupling terms depend on both the signs of the perturbations and the signal correlation between the endpoints. This suggests that impactful perturbations tend to involve intersecting edge pairs whose perturbation types are aligned with the signal correlation. The following remarks provide structural interpretations of impactful perturbations for $\mathbf{A}$ and $\mathbf{L}$.

**Remark 3.** *For the adjacency-based filter $g(\mathbf{S}) = \mathbf{A}$, impactful edge perturbations tend to have intersecting edge perturbations rather than distributed and disjoint edge perturbations. Those intersecting edge perturbations tend to have their perturbation types aligned with the second-order statistics of the non-intersecting vertices, ensuring that $\sigma_{uv}\sigma_{uv'}\mathbf{K}_{vv'} > 0$.*

**Remark 4.** *By definition, the function $\mathcal{R}(\cdot) : \mathcal{V} \times \mathcal{V} \to \mathbb{R}$ is a distance function in a metric space, and it follows the triangle inequality $\mathcal{R}(u, v) + \mathcal{R}(u, v') - \mathcal{R}(v, v') \geq 0$. Therefore, for $g(\mathbf{S}) = \mathbf{L}$, impactful edge perturbations tend to have intersecting edge perturbations consistent in type, either both additions or both deletions, i.e., $\sigma_{uv}\sigma_{uv'} = 1$.*

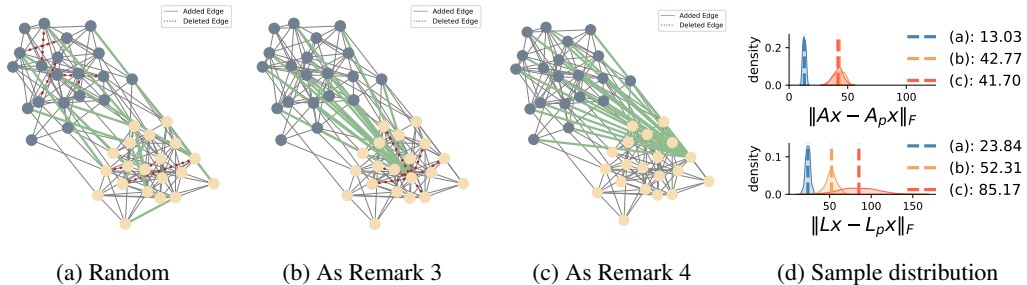

| (a) Random | (b) As Remark 3 | (c) As Remark 4 | (d) Sample distribution |

Figure 2: Edge perturbations on a graph and signals from the contextual stochastic block model.

To elaborate on these insights, we use the contextual stochastic block model (cSBM) as a demonstration. In cSBM [10], vertices within the same community have positively correlated signals, while signals between communities are negatively correlated. We adopt graph signal distribution defined in (37), with $\mathbf{K}_{uv} > 0$ if $u$ and $v$ belong to the same community, $\mathbf{K}_{uv} < 0$ otherwise. Figure 2 illustrates edge perturbations patterns constructed according to Remark 3 and Remark 4. We generate 100 graph signals sampled from (37) and plot the distribution of their embedding perturbations in Figure 2d. The results show that, for the same number of edge perturbations, the perturbation patterns following Remark 3 and Remark 4 produce the largest overall embedding perturbations for $\mathbf{A}$ and $\mathbf{L}$ separately, validating the structural intuitions described in both remarks.

Several prior studies [31, 46, 59] have shown that certain structural patterns of adversarial edge perturbations are more impactful than others. The key distinction is that: previous approaches design perturbations primarily with respect to the graph topology, such as vertex degree, while our study advocates for jointly considering both the graph topology and the signal distribution. In earlier work, Waniek et al. [50] proposed the heuristic adversarial perturbation method `DICE`, which disconnects nodes within the same community and connect nodes across communities. Our theoretical analysis, along with the cSBM case study, provides a rigorous justification for this heuristic by explaining from a perspective of embedding perturbation why and when this strategy is efficient. We note that our framework is broadly applicable, as it neither relies on a specific graph filter nor requires ground-truth labels from downstream tasks, thereby providing insights into the structure of impactful perturbations across diverse graph learning settings.

## 5  Empirical Study on Adversarial Edge Perturbation

Adversarial graph attacks are carefully designed, small changes applied to the graphs that can lead to unreliable model outputs. Understanding such adversarial vulnerabilities is crucial for assessing the robustness of graph learning models. In this section, we demonstrate the practical impact of our theoretical findings in adversarial graph attacks. Specifically, we develop a distribution-aware attack algorithm, `Prob-PGD`, to identify adversarial edge perturbations that significantly affect graph signal embeddings. Following our setting of inference-time stability of pre-trained models, we focus on *evasion* attacks on graphs (as opposed to *poisoning* attacks), where an adversary can modify only

edges of the graph, without changing the learned model parameters of graph filters and GCNNs. We consider an adversarial edge attack on undirected simple graphs and further assume that adversaries have a limited budget, allowing them to add or delete up to $m$ edges.

**Methodology.** We consider a task-agonistic edge attack where our goal of adversarial edge perturbation is to maximize the expected embedding perturbations from graph filters or $L$-layer GC-NNs

$$\mathcal{P}^* = \arg\max_{|\mathcal{P}|=m} \mathbb{E}[\|\mathbf{E}_g X\|_F^2], \quad \text{or} \quad \mathcal{P}^* = \arg\max_{|\mathcal{P}|=m} \mathbb{E}[\|X^{(L)} - X_p^{(L)}\|_F^2].$$

From Theorem 1 and Theorem 2, we characterize both embedding perturbations and establish their crucial dependence on the corresponding filter perturbation term $\langle \mathbf{K}, \mathbf{E}_g^T \mathbf{E}_g \rangle$. Based on these, we propose an adversarial edge perturbation design as

$$\hat{\mathcal{P}} = \arg\max_{|\mathcal{P}|=m} \langle \mathbf{K}, \mathbf{E}_g^T \mathbf{E}_g \rangle. \tag{8}$$

Here, the optimization objective involves the filter perturbation $\mathbf{E}_g$ and second-moment matrix $\mathbf{K}$. The filter perturbation $\mathbf{E}_g$ can be computed directly from its definition, $\mathbf{E}_g = g(\mathbf{S}) - g(\mathbf{S}_p)$, for any given filter and perturbation $\mathcal{P}$ (examples are provided in Appendix A.2). The second-moment matrix can be obtained using the empirical second-moment, i.e., $\mathbf{K}_{\text{samp}} = 1/N \sum_{i=1}^N \mathbf{x}_i \mathbf{x}_i^T$, with alternative approaches discussed in Appendix C.3. The main obstacle to solving this optimization problem is the computational cost, as the complexity of brute-force searching is $O(n^m)$. To circumvent this, we consider relaxing discrete constraints in the problem and applying a projected gradient descent method developed by Xu et al. [54] for efficient adversarial attack in `Prob-PGD`. We present the algorithmic details in Appendix C.1. We also refer the reader to Appendix C.8, where we discuss how the proposed method can be adapted to large-scale datasets and present experimental results.

**Baselines.** In our experiments, we compare our method `Prob-PGD` with two existing baselines: a randomize baseline, `Random`, which uniformly selects $m$ pair of vertices and alters their connectivity; another algorithm `Wst-PGD` from Kenlay et al. [25] is based on the same projected gradient descent approach but optimizes for the worst-case embedding metric (38). Other attack algorithms, such as those proposed in [54, 50], are not directly comparable, as they rely on specific loss functions, whereas our setting is task-agnostic.

### 5.1 Adversarial embedding attack

**Graph datasets.** We conduct experiments on a variety of graphs with different structural properties from both synthetic and real-world datasets. Specifically, we consider the following graph datasets: (1) Stochastic block model (SBM), which generates graphs with community structure [19]; (2) Barabasi-Albert model (BA), which generates random scale-free graphs through a preferential attachment mechanism [1]; (3) Watts-Strogatz model (WS) producing graphs with *small-world* properties [51]; (4) a random geometric (disc) graph model – Random Sensor Network; (5) Zachary's karate club network, a social network representing a university karate club [56]; (6) ENZYMES, consisting of protein tertiary structures obtained from the BRENDA enzyme database [41]. We refer to Appendix C.2 for detailed descriptions.

**Graph signals.** Among the above datasets, only the ENZYMES dataset contains real-world measurement signals associated with their vertices, while the remaining five datasets only contain graph structures without signals on their vertices. For these graph-only datasets, we generate 100 synthetic signals from an arbitrary distribution, such as random Gaussian signals, signals from cSBM, or the smoothness distribution (see Appendix C.2 for a detailed description). The second-moment matrix $\mathbf{K}$ is obtained from computing the sample graph signals, i.e, using $\mathbf{K}_{\text{samp}} = 1/100 \sum_{i=1}^{100} \mathbf{x}_i \mathbf{x}_i^T$.

**Results of graph filters embedding.** We evaluate adversarial edge perturbations on two representative filters: the graph adjacency (low-pass) and the graph Laplacian (high-pass), which capture distinct signal propagation behaviors in graph-based data processing. We apply different adversarial edge perturbation methods with a fixed budget of $m = 20$ edges (less than 2% of the maximum possible edges in most graphs). The results, shown in Figure 3, are presented as violin plots, illustrating the distribution of embedding perturbations across multiple graph signals—100 randomly sampled signals for synthetic datasets and 18 real-world measurement signals for ENZYMES. This provides a more detailed view of how edge perturbations affect different signal realizations. Across all datasets and filter types, our distribution-aware method `Prob-PGD` consistently induces the largest overall embedding perturbations, highlighting the effectiveness and robustness of our methodology.

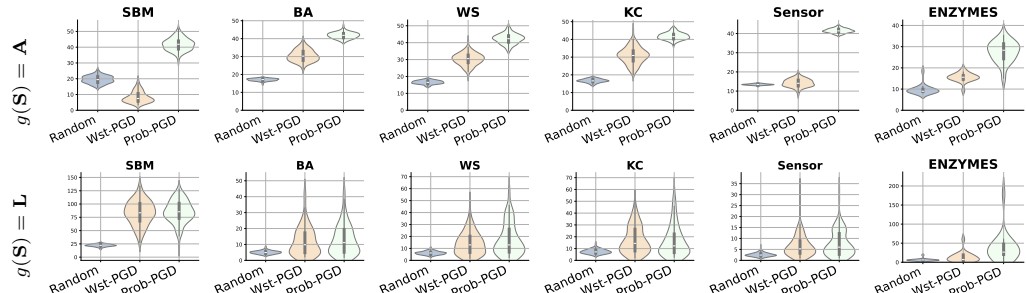

Figure 3: Violin plot of graph signal embedding perturbations for $g(\mathbf{S}) = \mathbf{A}$ (1st row) and $g(\mathbf{S}) = \mathbf{L}$ (2nd row). The y-axis shows sample-wise embedding perturbations i.e, $\|g(\mathbf{S})\mathbf{x}_i - g(\mathbf{S}_p)\mathbf{x}_i\|_2$ for signals $\{\mathbf{x}_i\}_{i=1}^d$ associated with each graph.

**Results of multilayer GCN embedding.** We conduct experiments on multilayer GCN architectures [27] using the normalized adjacency matrix $g(\mathbf{S}) = \tilde{\mathbf{D}}^{-1/2}\tilde{\mathbf{A}}\tilde{\mathbf{D}}^{-1/2}$. The tests are performed on contextual stochastic block models (cSBM) with 100 vertices, where each algorithm perturbs 20 edges. Since the learnable parameters $\{\mathbf{\Theta}^{(l)}\}_{l=1}^L$ also influence the final embeddings, we perform 200 repeated experiments with model weights independently resampled from a standard Gaussian distribution to ensure statistical reliability. Figure 4 presents the Frobenius norm of the embedding perturbations at each layer, visualized as box plots over the 200 trials. Across all layers and activation functions, our distribution-aware method `Prob-PGD` consistently outperforms the baselines, demonstrating the effectiveness of our probabilistic formulation in deeper architectures. We also observe that the performance gap between attack methods narrows in deeper layers, possibly due to the loosening of our theoretical bound with depth and inherent phenomena in deep GCNs, such as oversmoothing. Additional results with different model architectures are provided in Appendix C.5.

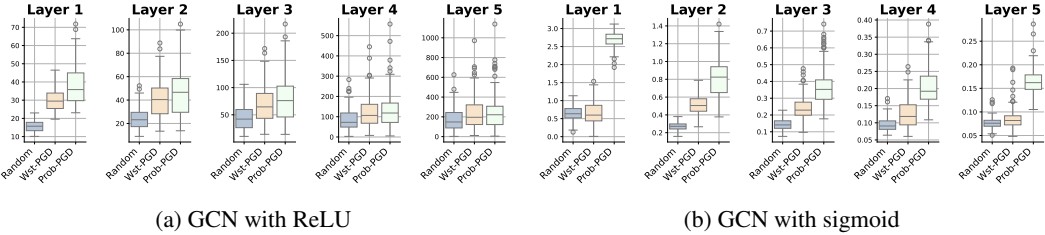

(a) GCN with ReLU  (b) GCN with sigmoid

Figure 4: Embedding perturbation of 5-layer GCNs with different activation function.

## 5.2 Adversarial attack on downstream tasks

We further evaluate how embedding-level adversarial perturbations affect the downstream performance of pre-trained GCNN models. Our experiments include three popular architectures: GCN [27], SGC [52], and CIN[3] [53]. More details about model architecture are provided in Table 4 in Appendix A.2, and training details are given in Appendix C.2.

**Graph classification.** We evaluate classification performance on two benchmark datasets: MUTAG (188 molecular graphs divided into two classes) and ENZYMES (600 protein graphs labeled by six enzyme classes). Both datasets provide numerical node features (e.g., atom types and physical measurements), from which we compute the second-moment matrix as $\mathbf{K}_{\text{samp}} = 1/d \sum_{i=1}^d \mathbf{x}_i \mathbf{x}_i^T$. For each dataset, we use 80% of the graphs to pre-train 10 GCNNs and keep their parameters fixed during evaluation. To assess their prediction robustness, we perturb $5\%$ edges in test set graphs using different attack algorithms and measure classification accuracy before and after perturbation.

**Node classification**. We use the Cora citation network (2708 vertices belonging to seven categories) with sparse bag-of-words node features. We pre-train 10 GCNNs on the unperturbed graphs with 140 labeled nodes and keep the model parameters fixed during evaluation. To assess the robustness

---

[3]We modify the standard GIN architecture by replacing the MLP in each layer with a single-layer perceptron, resulting in a simplified convolutional variant of GIN. Experiments (Appendix C.6) show that standard GINs with deeper MLPs exhibit a similar response to edge perturbations as their convolutional counterparts.

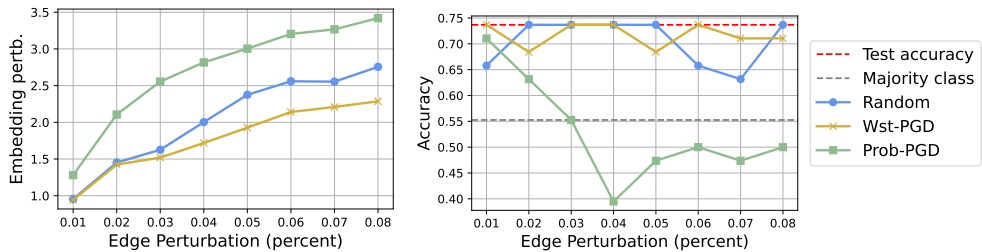

Figure 5: Classification performance of a pre-trained two-layer GCN on the MUTAG dataset under varying levels of edge perturbation.

of these pre-trained models, we apply different algorithms to perturb 1000 edges to the graph and evaluate classification accuracy on test set nodes before and after perturbation. Additional experiments on heterophilic datasets are provided in Appendix C.7.

Table 1: Prediction accuracy of GCNNs under edge perturbations produced by different methods.

| Dataset | Method | Prediction Accuracy(%) | | | Embedding Pert. $\| \cdot \|_F$ | | |
|---|---|---|---|---|---|---|---|
| | | GCN | GIN | SGC | GCN | GIN | SGC |
| MUTAG | Unpert. | $70.00_{\pm 2.68}$ | $82.89_{\pm 4.60}$ | $68.95_{\pm 1.58}$ | - | - | - |
| | Random | $73.16_{\pm 1.05}$ | $\mathbf{70.53}_{\pm 4.82}$ | $72.89_{\pm 1.21}$ | $2.24_{\pm 0.15}$ | $28.62_{\pm 2.34}$ | $3.57_{\pm 0.17}$ |
| | Wst-PGD | $69.21_{\pm 2.37}$ | $76.32_{\pm 8.15}$ | $68.16_{\pm 1.42}$ | $1.95_{\pm 0.13}$ | $37.47_{\pm 9.47}$ | $3.11_{\pm 0.16}$ |
| | Prob-PGD | $\mathbf{42.89}_{\pm 4.86}$ | $76.58_{\pm 8.02}$ | $\mathbf{41.58}_{\pm 3.87}$ | $\mathbf{3.09}_{\pm 0.16}$ | $\mathbf{96.26}_{\pm 11.39}$ | $\mathbf{5.21}_{\pm 0.25}$ |
| ENZYMES | Unpert. | $56.33_{\pm 3.10}$ | $56.75_{\pm 2.99}$ | $54.75_{\pm 3.48}$ | - | - | - |
| | Random | $49.58_{\pm 3.93}$ | $45.83_{\pm 2.64}$ | $45.50_{\pm 2.94}$ | $15.3_{\pm 0.7}$ | $81.3_{\pm 11.3}$ | $36.4_{\pm 1.2}$ |
| | Wst-PGD | $52.58_{\pm 3.56}$ | $39.75_{\pm 3.29}$ | $48.75_{\pm 4.27}$ | $11.5_{\pm 0.5}$ | $264.9_{\pm 34.7}$ | $27.2_{\pm 0.9}$ |
| | Prob-PGD | $\mathbf{44.25}_{\pm 2.09}$ | $\mathbf{36.75}_{\pm 4.62}$ | $\mathbf{40.75}_{\pm 3.83}$ | $\mathbf{17.9}_{\pm 0.9}$ | $\mathbf{297.9}_{\pm 57.1}$ | $\mathbf{43.6}_{\pm 1.6}$ |
| Cora | Unpert. | $80.06_{\pm 0.51}$ | $74.92_{\pm 1.86}$ | $80.33_{\pm 0.29}$ | - | - | - |
| | Random | $78.30_{\pm 0.82}$ | $71.75_{\pm 1.85}$ | $78.12_{\pm 0.64}$ | $93_{\pm 4}$ | $974_{\pm 496}$ | $136_{\pm 4}$ |
| | Wst-PGD | $78.47_{\pm 0.44}$ | $73.76_{\pm 2.07}$ | $78.30_{\pm 0.33}$ | $97_{\pm 7}$ | $16901_{\pm 12495}$ | $151_{\pm 5}$ |
| | Prob-PGD | $\mathbf{77.88}_{\pm 0.33}$ | $\mathbf{56.67}_{\pm 4.57}$ | $\mathbf{77.49}_{\pm 0.29}$ | $\mathbf{106}_{\pm 5}$ | $\mathbf{43565}_{\pm 33035}$ | $\mathbf{160}_{\pm 6}$ |

**Results.** We report classification accuracy and embedding perturbation magnitude (measured by the Frobenius norm) in Table 1, averaged over 10 pre-trained models. Across datasets and architectures, `Prob-PGD` consistently induces the largest embedding perturbations, which result in greater performance degradation in eight out of nine cases. Figure 5 further illustrates the performance of a pre-trained two-layer GCN on the MUTAG dataset under varying levels of edge perturbation. The left plot shows that `Prob-PGD` consistently induces the largest embedding perturbations across all perturbation levels. The right plot demonstrates that embedding-level perturbations transfer to downstream performance degradation. Notably, when the edge perturbation exceeds 3%, `Prob-PGD` reduces model accuracy to a level comparable to a trivial classifier that always predicts the majority class. Overall, our proposed algorithm `Prob-PGD`, although task-agnostic, still causes significant performance degradation in pre-trained GCNNs across different tasks. These results underscore the vulnerability of GCNNs to even small edge perturbations and reinforce the importance of comprehensive stability analysis in such settings.

## 6 Conclusion

In this paper, we propose a probabilistic framework for analyzing the embedding stability of GCNNs under edge perturbations. Unlike prior worst-case analyses, our formulation naturally enables a distribution-aware understanding of how graph structure and signal correlation jointly influence model sensitivity. This insight allows us to provide a structural interpretation of the importance of perturbation, and motivates the design of `Prob-PGD`, a new adversarial attack method shown to be effective through extensive experiments. While focused on inference-time stability, our study naturally supports broader robustness studies in graph machine learning, including adversarial training [33] by crafting adversarial examples. We leave such studies for future work.

## Acknowledgment

NZ acknowledges support from the Engineering and Physical Sciences Research Council (EPSRC) and IBM. XD acknowledges support from the Oxford-Man Institute of Quantitative Finance and EPSRC No. EP/T023333/1.

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

# A Summary on notations and concept definitions

## A.1 Summary of notations

Table 2: Summary on notations

| Notation | Definition |
| --- | --- |
| $\mathbf{x}$ | A deterministic graph signal, $\mathbf{x} \in \mathbb{R}^n$ |
| $X$ | Random graph signal (node feature) |
| $\mathcal{G}$ | Graph |
| $\mathcal{G}_p$ | Perturbed graph |
| $\mathbf{A}$ | Graph adjacency matrix |
| $\mathbf{A_p}$ | Adjacency matrix of the perturbed graph $\mathcal{G}_p$ |
| $\mathbf{S}$ | Graph shift operator, $\mathbf{S}_{uv} = 0$ if $\mathbf{A}_{uv} = 0$ |
| $\mathbf{S}_p$ | Graph shift operator on a perturbed graph, $(\mathbf{S}_p)_{uv} = 0$ if $(\mathbf{A}_p)_{uv} = 0$ |
| $g(\mathbf{S})$ | A graph filter defined on the shift operator $\mathbf{S}$ |
| $\mathbf{E_g}$ | Graph filter perturbation, $g(\mathbf{S}_p) - g(\mathbf{S}) = \mathbf{E_g}$ |
| $\mathbf{E}_g\mathbf{x}$ | Embedding perturbation for input $\mathbf{x}$, and $\mathbf{E}_g\mathbf{x} = g(\mathbf{S}_p)\mathbf{x} - g(\mathbf{S})\mathbf{x}$ |
| $\mathcal{D}$ | Distribution of graph signals |
| $\mathbf{K}$ | Second-moment matrix of graph signal from distribution $\mathcal{D}$ |
| $\mathbf{D}$ | Degree matrix, a diagonal matrix with entries $\mathbf{D}_{ii} = \sum_j \mathbf{A}_{ij}$ |
| $\mathbf{L}$ | Graph Laplacian where $\mathbf{L} = \mathbf{D} - \mathbf{A}$ |
| $\mathbf{D_p}$ | Degree matrix of the perturbed graph $\mathcal{G}_p$ |
| $\mathbf{L}_p$ | Graph Laplacian of perturbed graph $\mathcal{G}_p$, where $\mathbf{L}_p = \mathbf{D}_p - \mathbf{A}_p$ |
| $\mathbf{J}$ | All-one matrix |
| $\mathbf{I}$ | Identity matrix |
| $\tilde{\mathbf{A}}$ | Graph adjacency matrix with self-loop, $\tilde{\mathbf{A}} = \mathbf{A} + \mathbf{I}$ |
| $\tilde{\mathbf{D}}$ | Degree matrix with self-loop, $\tilde{\mathbf{D}}_{ii} = \sum_j \tilde{\mathbf{A}}_{ij}$ |
| $\mathcal{P}$ | Collection of perturbed vertex pairs, $\mathcal{P} = \{\{u, v\} \subset \mathcal{V} : \{u, v\}$ is perturbed$\}$ |
| $|\mathcal{P}|$ | Cardinality of the set $\mathcal{P}$ |
| $\|\mathbf{x}\|_2^2$ | quadratic norm of vector $\mathbf{x}$, where $\|\mathbf{x}\|_2^2 = \mathbf{x}\mathbf{x}^T$ |
| $\|\mathbf{M}\|$ | Spectral norm of matrix $\mathbf{M}$, where $\|\mathbf{M}\|$ is the largest singular value |
| $\|\mathbf{M}\|_F$ | Frobenius norm of matrix $\mathbf{M}$, where $\|\mathbf{M}\|_F = \sqrt{\sum_{ij} \mathbf{M}_{ij}^2}$ |
| $\langle \mathbf{A}, \mathbf{B} \rangle$ | Frobenius inner product, $\langle \mathbf{A}, \mathbf{B} \rangle = \sum_{ij} \mathbf{A}_{ij}\mathbf{B}_{ij}$ |
| $\mathbf{A} \odot \mathbf{B}$ | Hadamard product of two matrices, $(\mathbf{A} \odot \mathbf{B})_{ij} = \mathbf{A}_{ij}\mathbf{B}_{ij}$ |
| $\mathbf{A} \succeq 0$ | Matrix $\mathbf{A}$ is positive semidefinite |
| $\sigma(\cdot)$ | Non-linear activation function |
| $\mathbf{\Theta}^{(l)}$ | Learnable parameters in the $l$-th layer of a GCNN |

## A.2 Concept definitions

**Edge perturbation.** We take a simple graph $\mathcal{G}$, shown in Figure 6a, as an illustrative example. The graph signal on $\mathcal{G}$ is given by $\mathbf{x} = (0.1, 0.5, 0.9, 0.3, 0.7, 0.6)^T$. We perturb the graph $\mathcal{G}$ by deleting the edge between 4 and 6 and connecting 3 and 5, and the resulting perturbed graph is shown in Figure 6b. The edge perturbations can be represented as $\mathcal{P} = \{\{3, 5\}, \{4, 6\}\}$ with $\sigma_{46} = -1$ and $\sigma_{35} = 1$.

**Filter perturbation.** Edge perturbations in a graph induce corresponding changes in the graph filters. For example, when using the graph Laplacian as the filter, i.e., $g(\mathbf{S}) = \mathbf{L}$, the perturbation $\mathcal{P}$ results in a filter change given by

$$\mathbf{E_g} = \mathbf{L} - \mathbf{L}_p = \begin{pmatrix} 0 & 0 & 0 & 0 & 0 & 0 \\ 0 & 0 & 0 & 0 & 0 & 0 \\ 0 & 0 & -1 & 0 & 1 & 0 \\ 0 & 0 & 0 & 1 & 0 & -1 \\ 0 & 0 & 1 & 0 & -1 & 0 \\ 0 & 0 & 0 & -1 & 0 & 1 \end{pmatrix}.$$

Then the magnitude of embedding perturbation for graph signal $\mathbf{x}$ is $\|\mathbf{E}_g\mathbf{x}\|_2^2 = 0.26$.

Table 3: Common graph filters from [42, 40] and their perturbations

| Graph Filter $g(\mathbf{S})$ | Filter Perturbation $\mathbf{E}_g$ |
| --- | --- |
| polynomial of adjacency matrix: $\sum_{j=1}^{k} c_j \mathbf{A}$ | $\sum_{j=1}^{k} c_j(\mathbf{A} - \mathbf{A}_p)$ |
| polynomial of Laplacian: $\sum_{j=1}^{k} c_j \mathbf{L}$ | $\sum_{j=1}^{k} c_j(\mathbf{L} - \mathbf{L}_p)$ |
| low-pass filter: $(\mathbf{I} + \alpha\mathbf{L})^{-1}$ | $(\mathbf{I} + \alpha\mathbf{L})^{-1} - (\mathbf{I} + \alpha\mathbf{L}_p)^{-1}$ |
| heat diffusion filter: $e^{-\tau\mathbf{L}}$ | $e^{-\tau\mathbf{L}} - e^{-\tau\mathbf{L}_p}$ |

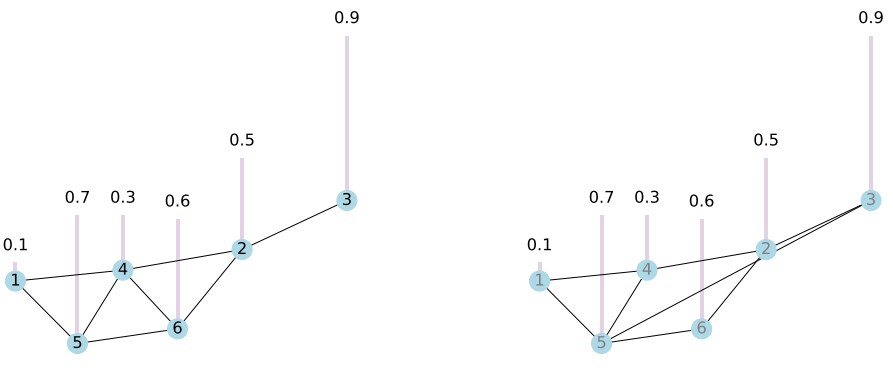

(a) graph $\mathcal{G}$ and associated signal $\mathbf{x}$      (b) perturbed graph $\mathcal{G}_p$ with $\mathbf{x}$

Figure 6: Attributed graph and graph perturbation.

For a general graph filter $g(\mathbf{S})$, the corresponding filter perturbation induced by $\mathcal{P}$ can be computed by first constructing $g(\mathbf{S})$ and $g(\mathbf{S}_p)$ from the original graph $\mathcal{G}$ and the perturbed graph $\mathcal{G}_p$ separately, and then taking their difference $\mathbf{E}_g = g(\mathbf{S}) - g(\mathbf{S}_p)$. We summarize in Table 3 some common filters and their perturbation.

## B  Proofs

### B.1  Proof of Theorem 1

**Theorem 1** (Stability of Graph Filters). *Let $X \in \mathbb{R}^n$ be a random graph signal from a distribution $\mathcal{D}$ with second moment matrix $\mathbf{K}$. Then for any graph filter perturbation $\mathbf{E}_g = g(\mathbf{S}) - g(\mathbf{S}_p)$, the expected change in output embedding is*

$$\mathbb{E}_{X \sim \mathcal{D}}[\|\mathbf{E}_g X\|_F^2] = \langle \mathbf{K}, \mathbf{E}_g^T \mathbf{E}_g \rangle. \tag{4}$$

*Proof.* We start with deriving the expression (4) for the expected embedding perturbation. Consider a random graph signal $X$ in $\mathbb{R}^n$, its $i$-th entry is a random variable, which we denote as $X_i$. Given a deterministic graph filter perturbation $\mathbf{E}_g$, the perturbation on the output embedding is $\mathbf{E}_g X$, which is also a random vector, and its squared Frobenius norm (Euclidean norm) can be written as

$$\|\mathbf{E}_g X\|_F^2 = X^T \mathbf{E}_g^T \mathbf{E}_g X = \sum_{i,j} X_i X_j (\mathbf{E}_g^T \mathbf{E}_g)_{ij}.$$

Then the expectation of this squared Euclidean norm of the embedding perturbation can be written as

$$\mathbb{E}_{X \sim D}[\|\mathbf{E}_g X\|_F^2] = \sum_{i,j} \mathbb{E}[X_i X_j](\mathbf{E}_g^T \mathbf{E}_g)_{ij} = \langle \mathbf{K}, \mathbf{E}_g^T \mathbf{E}_g \rangle,$$

where the last equality simply follows from the definition of the second-moment matrix $\mathbf{K} = \mathbb{E}[XX^T]$.

---

[4] We modify the standard GIN architecture by replacing the MLP in each layer with a single-layer perceptron, resulting in a simplified convolutional variant of GIN.

Table 4: GCNNs and their associated graph filters.

| GCNN Model | $g(\mathbf{S})$ | $\mathbf{E}_g$ | $C$ from (A1) |
|---|---|---|---|
| GCN [27] | $\tilde{\mathbf{D}}^{-1/2}\tilde{\mathbf{A}}\tilde{\mathbf{D}}^{-1/2}$ | $\tilde{\mathbf{D}}^{-1/2}\tilde{\mathbf{A}}\tilde{\mathbf{D}}^{-1/2} - \tilde{\mathbf{D}}_p^{-1/2}\tilde{\mathbf{A}}_p\tilde{\mathbf{D}}_p^{-1/2}$ | 1 |
| GIN [53][4] | $(1+\epsilon^{(k)})\mathbf{I} + \mathbf{A}$ | $\mathbf{A} - \mathbf{A}_p$ | max degree |
| SGC($k$) [52] | $\left(\tilde{\mathbf{D}}^{-1/2}\tilde{\mathbf{A}}\tilde{\mathbf{D}}^{-1/2}\right)^k$ | $\left(\tilde{\mathbf{D}}^{-1/2}\tilde{\mathbf{A}}\tilde{\mathbf{D}}^{-1/2}\right)^k - \left(\tilde{\mathbf{D}}_p^{-1/2}\tilde{\mathbf{A}}_p\tilde{\mathbf{D}}_p^{-1/2}\right)^k$ | 1 |

Next, we show the concentration of the embedding perturbation. Using Markov's inequality, we have that, for the non-negative random variable $\|\mathbf{E}_g X\|_2^2$ and any $c > 0$

$$\mathbb{P}_{X\sim\mathcal{D}}\left(\|\mathbf{E}_g X\|_2^2 \geq (1+c)\mathbb{E}[\|\mathbf{E}_g X\|_2^2]\right) \leq \frac{1}{1+c}.$$

$\square$

### B.2 Proof of Corollary 4

The following Lemma contains useful inequalities that we will use in the proofs.

**Lemma 1** (Basic matrix inequalities). *For any $\mathbf{A}, \mathbf{B} \in \mathbb{R}^{n\times n}$, we have*

$$\lambda_{\min}^2(\mathbf{A})\|\mathbf{B}\|_F^2 \leq \|\mathbf{A}\mathbf{B}\|_F^2 \leq \lambda_{\max}^2(\mathbf{A})\|\mathbf{B}\|_F^2 \tag{9}$$

*For any $\mathbf{A}, \mathbf{B} \succeq 0$, we have*

$$\langle \mathbf{A}, \mathbf{B}\rangle \leq \|A\|\mathrm{Tr}(\mathbf{B}) \tag{10}$$

*For any $\mathbf{A} \in \mathbb{R}^{n\times m}\mathbf{B} \in \mathbb{R}^{m\times d} \succeq 0$, we have*

$$\|\mathbf{A}\mathbf{B}\|_F^2 \leq \|\mathbf{A}\|^2\|\mathbf{B}\|_F^2, \tag{11}$$

*where $\|\mathbf{A}\|^2$ is the larges singular value of $\mathbf{A}$.*

*Proof.* The inequalities from (9) are adapted from Theorem 1 in [47].

To prove the inequality (10), we denote the eigendecomposition $\mathbf{A} = \mathbf{U}\mathbf{\Lambda}\mathbf{U}^T$ and $\mathbf{B} = \mathbf{V}\mathbf{\Sigma}\mathbf{V}^T$, where $\mathbf{\Lambda} = \mathrm{diag}(\lambda_i)$ and $\mathbf{\Sigma} = \mathrm{diag}(\sigma_i)$. Then we have

$$\begin{aligned}
\langle \mathbf{A}, \mathbf{B}\rangle &= \mathrm{Tr}(\mathbf{U}\mathbf{\Lambda}\mathbf{U}^T\mathbf{V}\mathbf{\Sigma}\mathbf{V}^T)\\
&= \mathrm{Tr}(\mathbf{\Lambda}\mathbf{U}^T\mathbf{V}\mathbf{\Sigma}\mathbf{V}^T\mathbf{U})\\
&= \sum_{i=1}^n \lambda_i(\mathbf{U}^T\mathbf{V}\mathbf{\Sigma}\mathbf{V}^T\mathbf{U})_{ii}\\
&\leq \|\mathbf{A}\|\sum_{i=1}^n(\mathbf{U}^T\mathbf{V}\mathbf{\Sigma}\mathbf{V}^T\mathbf{U})_{ii}\\
&= \|\mathbf{A}\|\mathrm{Tr}(\mathbf{U}^T\mathbf{V}\mathbf{\Sigma}\mathbf{V}^T\mathbf{U}) = \|\mathbf{A}\|\mathrm{Tr}(\mathbf{B})
\end{aligned}$$

Here the inequality follows because $\lambda_i \geq 0$ and $\sigma_i \geq 0$ for all $i$.

To prove the inequality (11), note that

$$\|\mathbf{A}\mathbf{B}\|_F^2 = \mathrm{Tr}(\mathbf{B}^T\mathbf{A}^T\mathbf{A}\mathbf{B}) = \langle \mathbf{B}\mathbf{B}^T, \mathbf{A}^T\mathbf{A}\rangle.$$

Therefore, (10) directly implies (11). $\square$

### B.3 Proof of Corollary 2

**Corollary 2** (Stability of Single-Layer GCNNs). *Let $X \in \mathbb{R}^{n\times d}$ be a random matrix whose columns are i.i.d. graph signals from a distribution $\mathcal{D}$ over $\mathbb{R}^n$, with second-moment matrix $\mathbf{K}$, and $\sigma(\cdot)$ be $C_\sigma$-Lipschitz continuous. Then the expected embedding perturbation of a single-layer GCNN satisfies*

$$\mathbb{E}_{X\sim\mathcal{D}}[\|X^{(1)} - X_p^{(1)}\|_F^2] \leq dC_\sigma^2\|\mathbf{\Theta}^{(1)}\|^2\langle \mathbf{K}, \mathbf{E}_g^T\mathbf{E}_g\rangle.$$

*Proof.* Under edge perturbations, the induced output embedding perturbation of a single-layered GCNN is more complicated than outputs of graph filters due to the existence of the non-linear activation function and the learned weight matrix $\mathbf{\Theta}^{(1)}$.

To handle the non-linearity, we consider upper bound the change in output in the following way:

$$
\begin{aligned}
X^{(1)} - X_p^{(1)} &= \sigma\left(g(\mathbf{S})X\mathbf{\Theta}^{(1)}\right) - \sigma\left(g(\mathbf{S}_p)X\mathbf{\Theta}^{(1)}\right) \\
&\leq C_\sigma\left|g(\mathbf{S})X\mathbf{\Theta}^{(1)} - g(\mathbf{S}_p)X\mathbf{\Theta}^{(1)}\right| \\
&= C_\sigma|\mathbf{E}_g X\mathbf{\Theta}^{(1)}|.
\end{aligned}
\tag{12}
$$

In (12), the inequality denotes entrywise less than, and $|\cdot|$ denotes entrywise absolute value on the matrix. The inequality (12) follows from the Lipschitz continuity of the activation function $\sigma(\cdot)$.

Furthermore, to handle $\mathbf{\Theta}^{(1)}$, we apply a matrix inequality (11) from Lemma 1 have that

$$
\|\mathbf{E}_g X\mathbf{\Theta}^{(1)}\|_F^2 = \|\mathbf{\Theta}^{(1)}\|^2\|\mathbf{E}_g X\|_F^2,
\tag{13}
$$

where $\|\mathbf{\Theta}^{(1)}\|$ denotes the larges singular value of $\mathbf{\Theta}^{(1)}$. Combining (12) and (13), we have the magnitude of the embedding perturbation

$$
\begin{aligned}
\mathbb{E}[\|X^{(1)} - X_p^{(1)}\|_F^2] &\leq C_\sigma^2 \mathbb{E}[\|\mathbf{E}_g X\mathbf{\Theta}^{(1)}\|_F^2] \\
&= C_\sigma^2\|\mathbf{\Theta}^{(1)}\|^2 \mathbb{E}[\|\mathbf{E}_g X\|_F^2] \\
&= dC_\sigma^2\|\mathbf{\Theta}^{(1)}\|^2\langle\mathbf{K}, \mathbf{E}_g^T\mathbf{E}_g\rangle,
\end{aligned}
\tag{14}
$$
$$
\tag{15}
$$

where (14) follows from (13) and (15) follows Theorem 1. $\qquad\square$

## B.4 Proof of Theorem 2

**Theorem 2** (Stability of $L$-layer GCNN). *Let $X \in \mathbb{R}^{n \times d}$ be a random matrix whose columns are i.i.d. graph signals from a distribution $\mathcal{D}$ over $\mathbb{R}^n$, with second-moment matrix $\mathbf{K}$. Consider an $L$-layer GCNN built on a graph filter $g(\mathbf{S})$ and let the corresponding filter perturbation be $\mathbf{E}_g = g(\mathbf{S}) - g(\mathbf{S}p)$. Suppose the model satisfies assumptions (A1) (A2) and (A3). Then the expected embedding perturbation of the L-layer GCNN satisfies*

$$
\mathbb{E}_{X\sim\mathcal{D}}\left[\|\mathbf{X}^{(L)} - \mathbf{X}_p^{(L)}\|_F^2\right] \leq d\,C_\sigma^{2L}C^{2L-2}\underbrace{\prod_{j=1}^{L}\|\mathbf{\Theta}^{(j)}\|^2}_{model}\left((L-1)\underbrace{\|\mathbf{E}_g\|^2}_{\substack{model \\ pert.}}\underbrace{\mathrm{tr}(\mathbf{K})}_{signals} + \underbrace{\langle\mathbf{K}, \mathbf{E}_g^T\mathbf{E}_g\rangle}_{\text{K-modulated pert.}}\right).
\tag{5}
$$

*Proof.* To establish our result, we first observe that if the following

$$
\|X^{(L)} - X_p^{(L)}\|_F^2 \leq C_\sigma^{2L}C^{2L-2}\prod_{j=1}^{L}\|\mathbf{\Theta}^{(l)}\|^2((L-1)\|\mathbf{E}_g\|^2\|X\|_F^2 + \|\mathbf{E}_g X\|_F^2)
\tag{16}
$$

holds with probability 1. Then, by Theorem 1, we can easily show

$$
\mathbb{E}[\|X^{(L)} - X_p^{(L)}\|_F^2] \leq dC_\sigma^{2L}C^{2L-2}\prod_{j=1}^{L}\|\mathbf{\Theta}^{(l)}\|^2((L-1)\|\mathbf{E}_g\|^2\mathrm{Tr}(\mathbf{K}) + \langle\mathbf{K}, \mathbf{E}_g^T\mathbf{E}_g\rangle).
\tag{17}
$$

Thus, it remains to show that condition (16) holds, which we prove by induction.

**Base case** ($l = 1$)**:** In previous proof of Corollary 2, equation (12) and (13) imply it holds with probability 1 that

$$
\|X^{(1)} - X_p^{(1)}\|_F^2 \leq C_\sigma^2\|\mathbf{\Theta}^{(1)}\|^2\|\mathbf{E}_g X\|_F^2.
$$

**Induction step :** For $1 < l < L$, assume that for the $l$-th layer embedding, it has

$$
\|X^{(l)} - X_p^{(l)}\|_F^2 = C_\sigma^{2l}C^{2l-2}\prod_{j=1}^{l}\|\mathbf{\Theta}^{(j)}\|^2((l-1)\|\mathbf{E}_g\|^2\|X\|_F^2 + \|\mathbf{E}_g X\|_F^2).
\tag{18}
$$

Then, for the $(l + 1)$-th layer, its embedding perturbation can be first simplified as follows

$$\|X^{(l+1)} - X_p^{(l+1)}\|_F^2 = \|\sigma^{(l+1)}(g(\mathbf{S})X^l\mathbf{\Theta}^{(l+1)}) - \sigma^{(l+1)}(g(\mathbf{S}_p)X_p^l\mathbf{\Theta}^{(l+1)})\|_F^2$$

$$\leq C_\sigma^2\|g(\mathbf{S})X^l\mathbf{\Theta}^{(l+1)} - g(\mathbf{S}_p)X_p^l\mathbf{\Theta}^{(l+1)}\|_F^2 \tag{19}$$

$$\leq C_\sigma^2\|\mathbf{\Theta}^{(l+1)}\|^2\|g(\mathbf{S})X^{(l)} - g(\mathbf{S}_p)X_p^{(l)}\|_F^2 \tag{20}$$

Here (19) follows form the Lipschitz continuity of $\sigma^{(l+1)}(\cdot)$, and (20) follows Frobenius-spectral inequality (11) from Lemma 1. The term $\|g(\mathbf{S})X^{(l)} - g(\mathbf{S}_p)X_p^{(l)}\|_F^2$ contains two sources of perturbation: perturbation in the graph filter $g(\mathbf{S})$ and propagated signal perturbation from pervious layer $X_p^{(l)}$. We decompose them as follows

$$\|g(\mathbf{S})X^{(l)} - g(\mathbf{S}_p)X_p^{(l)}\|_2^2 = \|g(\mathbf{S})X^{(l)} - g(\mathbf{S}_p)X^{(l)} + g(\mathbf{S}_p)X^{(l)} - g(\mathbf{S}_p)X_p^{(l)}\|_F^2$$

$$\leq \|g(\mathbf{S})X^{(l)} - g(\mathbf{S}_p)X^{(l)}\|_F^2 + \|g(\mathbf{S}_p)X^{(l)} - g(\mathbf{S}_p)X_p^{(l)}\|_F^2$$

$$\leq \|\mathbf{E}_gX^{(l)}\|_F^2 + \|g(\mathbf{S}_p)\|^2\|X^{(l)} - X_p^{(l)}\|_F^2 \tag{21}$$

$$\leq \|\mathbf{E}_gX^{(l)}\|_F^2 + C^2\|X^{(l)} - X_p^{(l)}\|_F^2. \tag{22}$$

Here (21) follows from the Frobenius-spectral inequality (11) in Lemma 1, and (22) holds by the assumption $\|g(\mathbf{S}_p)\| \leq C$. Here in (22), the first term captures the change of embedding with input $X^{(l)}$ caused by the filter perturbation, and the second term accounts for the embedding change from the $l$-th layer, which is given by the inductive hypothesis in (18). Therefore, we only need to further bound the term $\|\mathbf{E}_gX^{(l)}\|_F^2$. By definition of the GCNN, $X^{(l)}$ is recursively computed following

$$X^{(j)} = \sigma^{(j)}\left(g(\mathbf{S})X^{(j-1)}\mathbf{\Theta}^{(j)}\right) \quad \text{for } j = 1, \ldots, l.$$

Becasue of Assumption (A2) and (A3), we further have that $\sigma^{(j)}\left(g(\mathbf{S})X^{(j-1)}\mathbf{\Theta}^{(j)}\right)$ is entrywise upper bounded by $|C_\sigma|\left|g(\mathbf{S})X^{(j-1)}\mathbf{\Theta}^{(j)}\right|$. Therefore, we have

$$\|\mathbf{E}_gX^{(l)}\|_F^2 \leq C_\sigma^2\|\mathbf{E}_gg(\mathbf{S})X^{(l-1)}\mathbf{\Theta}^{(l)}\|_F^2 \leq C_\sigma^2C^2\|\mathbf{\Theta}^{(l)}\|^2\|\mathbf{E}_g\|^2\|X^{(l-1)}\|_F^2 \tag{23}$$

The second inequality in (23) is obtained by using the Frobenius-spectral inequality from Lemma 1. Therefore, we can also recursively bound the squared Frobenius norm of $\mathbf{E}_gX^{(l)}$ and obtain

$$\|\mathbf{E}_gX^{(l)}\|_F^2 \leq C_\sigma^{2l}C^{2l}\prod_{j=1}^l\|\mathbf{\Theta}^{(j)}\|^2\|\mathbf{E}_g\|^2\|X\|_F^2. \tag{24}$$

Combining the above results and the inductive hypothesis (18), we have

$$\|X^{(l+1)} - X_p^{(l+1)}\|_F^2$$

$$\leq C_\sigma^2\|\mathbf{\Theta}^{(l+1)}\|^2\|g(\mathbf{S})X^{(l)} - g(\mathbf{S}_p)X_p^{(l)}\|_F^2 \tag{25}$$

$$\leq C_\sigma^2\|\mathbf{\Theta}^{(l+1)}\|^2(\|\mathbf{E}_gX^{(l)}\|_F^2 + C^2\|X^{(l)} - X_p^{(l)}\|_F^2) \tag{26}$$

$$\leq C_\sigma^2\|\mathbf{\Theta}^{(l+1)}\|^2\left(C_\sigma^{2l}C^{2l}\prod_{j=1}^l\|\mathbf{\Theta}^{(j)}\|^2\|\mathbf{E}_g\|^2\|X\|_F^2 + C^2\|X^{(l)} - X_p^{(l)}\|_F^2\right) \tag{27}$$

$$\leq C_\sigma^{2l+2}C^{2l}\prod_{j=1}^{l+1}\|\mathbf{\Theta}^{(j)}\|^2(l\|\mathbf{E}_g\|^2\|X\|_F^2 + \|\mathbf{E}_gX\|_F^2). \tag{28}$$

Here, equation (25) follows from (20); inequality (26) is derived from (22); inequality (27) follows from (24); and the final step, equation (28), follows from the inductive hypothesis (18). □

## B.5 Proof of Proposition 1

**Proposition 1.** *Consider an edge perturbation set $\mathcal{P}$. If the graph filter is the adjacency matrix, i.e., $g(\mathbf{S}) = \mathbf{A}$, then the expected embedding perturbation satisfies*

$$\mathbb{E}_{X\sim\mathcal{D}}[\|\mathbf{E}_gX\|_2^2] = \langle\mathbf{K}, \mathbf{E}_g^T\mathbf{E}_g\rangle = \sum_{\{u,v\}\in\mathcal{P}}(\mathbf{K}_{uu} + \mathbf{K}_{vv}) + 2\sum_{\{u,v\},\{u,v'\}\in\mathcal{P}}\sigma_{uv}\sigma_{uv'}\mathbf{K}_{vv'}. \tag{6}$$

*If the graph filter is the graph Laplacian, i.e., $g(\mathbf{S}) = \mathbf{L}$*

$$\mathbb{E}_{X \sim \mathcal{D}}[\|\mathbf{E}_g X\|_2^2] = \langle \mathbf{K}, \mathbf{E}_g^T \mathbf{E}_g \rangle = 2 \sum_{\{u,v\} \in \mathcal{P}} \mathcal{R}(u,v) + \sum_{\{u,v\},\{u,v'\} \in \mathcal{P}} \sigma_{uv} \sigma_{uv'} (\mathcal{R}(u,v) + \mathcal{R}(u,v') - \mathcal{R}(v,v')),$$

$$(7)$$

*where $\mathcal{R}(u,v)$ is defined as $\mathcal{R}(u,v) \triangleq \mathbb{E}[(X_u - X_v)^2]$.*

*Proof.* (Case 1.) Proof for adjacency filter $g(\mathbf{S}) = \mathbf{A}$.
We denote $\mathbb{I}_{uv}$ the edge indicator matrix, where the entries of $\mathbb{I}_{uv}$ is all 0 other than entry $(u,v)$ and $(u,v)$ have value 1. Then, for a edge perturbation $\mathcal{P}$, we can decompose the filter perturbation as

$$\mathbf{E}_g = \mathbf{A}_p - \mathbf{A} = \sum_{\{u,v\} \in \mathcal{P}} \sigma_{uv} \mathbb{I}_{uv},$$

where $\sigma_{uv} = \pm 1$ indicating adding or deleting an edge between $u$ and $v$. Then, we have

$$(\mathbf{A}_p - \mathbf{A})^T (\mathbf{A}_p - \mathbf{A}) = \left( \sum_{\{u,v\} \in \mathcal{P}} \sigma_{uv} \mathbb{I}_{uv} \right) \left( \sum_{\{u,v\} \in \mathcal{P}} \sigma_{uv} \mathbb{I}_{uv} \right)$$

$$= \sum_{\{u,v\} \in \mathcal{P}} \mathbb{I}_{uv}^2 + \sum_{\{u,v\},\{u,v'\} \in \mathcal{P}} \sigma_{uv} \sigma_{uv'} (\mathbb{I}_{uv} \mathbb{I}_{uv'} + \mathbb{I}_{uv'} \mathbb{I}_{uv}) + \sum_{\{u,v\},\{u,v'\} \in \mathcal{P}} \sigma_{uv} \sigma_{v'u} (\mathbb{I}_{uv} \mathbb{I}_{v'u} + \mathbb{I}_{v'u} \mathbb{I}_{uv})$$

$$= \sum_{\{u,v\} \in \mathcal{P}} \mathbb{I}_{uu} + \mathbb{I}_{vv} + \sum_{\{u,v\},\{u,v'\} \in \mathcal{P}} \sigma_{uv} \sigma_{uv'} (\mathbb{I}_{vv'} + \mathbb{I}_{v'v}) + \sum_{\{u,v\},\{u,v'\} \in \mathcal{P}} \sigma_{uv} \sigma_{uv'} (\mathbb{I}_{vv'} + \mathbb{I}_{v'v}).$$

$$(29)$$

Therefore, we can further decompose the expected embedding perturbation from Theorem 1 as

$$\mathbb{E}[\|\mathbf{E}_g X\|_2^2] = \langle \mathbf{E}_g^T \mathbf{E}_g, \mathbf{K} \rangle = \langle (\mathbf{A}_p - \mathbf{A})^T (\mathbf{A}_p - \mathbf{A}), \mathbf{K} \rangle$$

$$= \sum_{\{u,v\} \in \mathcal{P}} \langle \mathbb{I}_{uu} + \mathbb{I}_{vv}, \mathbf{K} \rangle + \sum_{\{u,v\},\{u,v'\} \in \mathcal{P}} \sigma_{uv} \sigma_{uv'} \langle \mathbb{I}_{vv'} + \mathbb{I}_{v'v}, \mathbf{K} \rangle + \sum_{\{u,v\},\{u,v'\} \in \mathcal{P}} \sigma_{uv} \sigma_{uv'} \langle \mathbb{I}_{vv'} + \mathbb{I}_{v'v}, \mathbf{K} \rangle$$

$$(30)$$

$$= \sum_{\{u,v\} \in \mathcal{P}} \mathbf{K}_{uu} + \mathbf{K}_{vv} + 2 \sum_{\{u,v\},\{u,v'\} \in \mathcal{P}} \sigma_{uv} \sigma_{uv'} \mathbf{K}_{vv'} + 2 \sum_{\{u,v\},\{u,v'\} \in \mathcal{P}} \sigma_{uv} \sigma_{uv'} \mathbf{K}_{vv'}. \qquad (31)$$

Here (30) follows from (29). From (31), we have that for each pair of perturbed edges $\{u,v\} \in \mathcal{P}$, its contribution to the overall embedding perturbation comes from two parts: the unitary term $\mathbf{K}_{uu} + \mathbf{K}_{vv} = \mathbb{E}[X_u^2 + X_v^2]$; and the coupling term, which further add $2\sigma_{uv}\sigma_{uv'}\mathbf{K}_{vv'} = 2\sigma_{uv}\sigma_{uv'}\mathbb{E}[X_v X_{v'}]$ for each of other perturbed edges $\{u,v'\} \in \mathcal{P}$ that intersects with $\{u,v\}$.

(Case 2.) Proof for Laplacian filter $g(\mathbf{S}) = \mathbf{L}$
To start, we introduce the oriented edge indicator vector $\mathbf{b}_{uv} \in \{1, -1, 0\}^n$, which only has $\pm 1$ on entries that correspond to two vertices $u$ and $v$ connected by an edge and zero elsewhere. For a directed graph,

$$\mathbf{b}_{uv}(i) = \begin{cases} 1 & \text{if the edge points to } i \\ -1 & \text{if the edge leaves } i \\ 0 & \text{o.w.} \end{cases}$$

This definition can be extended to undirected graphs with arbitrary directions assigned to each edge. With the definition of edge indicator vector, we can then decompose the Laplacian matrix as

$$\mathbf{L} = \sum_{\{u,v\} \in \mathcal{E}} \mathbf{b}_{uv} \mathbf{b}_{uv}^T.$$

This expression allows us to decompose the filter perturbation to perturbation induced by each edge

$$\mathbf{E}_g = \mathbf{L}_p - \mathbf{L} = \sum_{\{u,v\} \in \mathcal{P}} \sigma_{uv} \mathbf{b}_{uv} \mathbf{b}_{uv}^T. \qquad (32)$$

Here $\mathcal{P}$ denote the collection of vertex pairs that are perturbed. For $\{u, v\} \in \mathcal{P}$, we denote $\sigma_{uv} = 1$ if an edge is added to connect $u$ and $v$ and $\sigma_{uv} = -1$ if the edge between $u$ and $v$ is deleted. Then, we can further derive

$$
\begin{aligned}
(\mathbf{L}_p - \mathbf{L})^T(\mathbf{L}_p - \mathbf{L}) &= \left( \sum_{\{u,v\} \in \mathcal{P}} \sigma_{uv} \mathbf{b}_{uv} \mathbf{b}_{uv}^T \right) \left( \sum_{\{u,v\} \in \mathcal{P}} \sigma_{uv} \mathbf{b}_{uv} \mathbf{b}_{uv}^T \right) \\
&= \sum_{\{u,v\} \in \mathcal{P}} \mathbf{b}_{uv} \mathbf{b}_{uv}^T \mathbf{b}_{uv} \mathbf{b}_{uv}^T + \sum_{\{u,v\},\{u,v'\} \in \mathcal{P}} \sigma_{uv} \sigma_{uv'} \mathbf{b}_{uv} \mathbf{b}_{uv}^T \mathbf{b}_{uv'} \mathbf{b}_{uv'}^T + \sum_{\{u,v\},\{u,v'\} \in \mathcal{P}} \sigma_{uv} \sigma_{v'u} \mathbf{b}_{uv} \mathbf{b}_{uv}^T \mathbf{b}_{v'u} \mathbf{b}_{v'u}^T \\
&= 2 \sum_{\{u,v\} \in \mathcal{P}} (\mathbb{1}_{uu} + \mathbb{1}_{vv} - \mathbb{1}_{uv} - \mathbb{1}_{vu}) + \sum_{\{u,v\},\{u,v'\} \in \mathcal{P}} \sigma_{uv} \sigma_{uv'} (2\mathbb{1}_{uu} + \mathbb{1}_{vv'} + \mathbb{1}_{v'v} - \mathbb{1}_{uv'} - \mathbb{1}_{v'u} - \mathbb{1}_{vu} - \mathbb{1}_{uv}) \\
&\quad + \sum_{\{u,v\},(v',u) \in \mathcal{P}} \sigma_{uv} \sigma_{v'u} (2\mathbb{1}_{uu} + \mathbb{1}_{vv'} + \mathbb{1}_{v'v} - \mathbb{1}_{uv'} - \mathbb{1}_{v'u} - \mathbb{1}_{vu} - \mathbb{1}_{uv})
\end{aligned}
\tag{33}
$$

Combining the above perturbation decomposition and Theorem 1, we have the following decomposition on the expected embedding perturbation

$$
\begin{aligned}
\mathbb{E}[\|\mathbf{E}_g X\|_2^2] &= \langle \mathbf{E}_g^T \mathbf{E}_g, \mathbf{K} \rangle = \langle (\mathbf{L}_p - \mathbf{L})^T (\mathbf{L}_p - \mathbf{L}), \mathbf{K} \rangle \\
&= 2 \sum_{\{u,v\} \in \mathcal{P}} (\mathbf{K}_{uu} + \mathbf{K}_{vv} - \mathbf{K}_{uv} - \mathbf{K}_{vu}) + 2 \sum_{\{u,v\},\{u,v'\} \in \mathcal{P}} \sigma_{uv} \sigma_{uv'} (\mathbf{K}_{uu} + \mathbf{K}_{vv'} - \mathbf{K}_{uv'} - \mathbf{K}_{vu}) \quad (34) \\
&= 2 \sum_{\{u,v\} \in \mathcal{P}} \mathcal{R}\{u,v\} + \sum_{\{u,v\},\{u,v'\} \in \mathcal{P}} \sigma_{uv} \sigma_{uv'} \mathcal{R}(u,v) + \mathcal{R}(u,v') - \mathcal{R}(v,v'). \quad (35)
\end{aligned}
$$

Here, (34) follows from (33). In (35), we define $\mathcal{R}(u,v) \triangleq \mathbb{E}[(X_u - X_v)^2]$. $\qquad \square$

## C  Experimental details

### C.1  Algorithm: `Prob-PGD`

As we discussed, attacking graph filters and GCNNs by optimizing the second-moment matrix modulated filter perturbation term $\langle \mathbf{K}, \mathbf{E}_g^T \mathbf{E}_g \rangle$ is computationally expensive due to the combinatorial nature of the problem. To circumvent this, we consider relaxing discrete constraints in the problem and applying a projected gradient descent method for efficient adversarial attack. This strategy is adapted from a prior work [54].

We denote $\mathbb{1}_{\mathcal{P}}$ the perturbation indicator matrix in $\{0,1\}^{n \times n}$, where the entry $(\mathbb{1}_{\mathcal{P}})_{uv} = (\mathbb{1}_{\mathcal{P}})_{vu} = 1$ if $\{u,v\} \in \mathcal{P}$ and $(\mathbb{1}_{\mathcal{P}})_{uv} = 0$ otherwise. For notational clarity, we denote the objective function, the expected filter embedding perturbation, as $f(\mathbb{1}_{\mathcal{P}}) = \langle \mathbf{K}, \mathbf{E}_g^T \mathbf{E}_g \rangle$. This function $f(\mathbb{1}_{\mathcal{P}})$ is determined by the prespecified filter function $g(\mathbf{S})$, the signal second-moment matrix $\mathbf{K}$ and a given graph perturbation $\mathcal{P}$. With a budget of perturbing at most $m$ edges, the optimal adversarial edge attack can be formulated as finding a maximizer of the following problem

$$
\begin{aligned}
\max \quad & f(\mathbb{1}_{\mathcal{P}}) \\
s.t. \quad & \mathbb{1}_{\mathcal{P}} \in \{0,1\}^{n \times n} \\
& \langle \mathbb{1}_{\mathcal{P}}, \mathbf{J} \rangle \leq 2m
\end{aligned}
\tag{P1}
$$

The hardness of solving (P1) mostly comes from the integer constraint, therefore, we consider relaxing this constraint to the following

$$
\begin{aligned}
\max \quad & f(\mathbf{M}) \\
s.t. \quad & \forall u, v, \mathbf{M}_{uv} \in [0,1] \\
& \langle \mathbf{M}, \mathbf{J} \rangle \leq 2m
\end{aligned}
\tag{P2}
$$

The relaxed problem (P2) has a convex feasible region and continuous objective function. Moreover, according to [54] (Proposition 1) for a general matrix $\mathbf{M}$, projecting it to the feasible region of (P2) is simply

$$
\Pi(\mathbf{M}) = \begin{cases} P_{[0,1]}[\mathbf{M} - \mu \mathbf{J}] & \text{if } \mu > 0 \text{ and } \langle \mathbf{J}, \mathbf{M} \rangle = 2m \\ P_{[0,1]}[\mathbf{M}] & \text{if } \langle \mathbf{J}, \mathbf{M} \rangle \leq 2m. \end{cases}
\tag{36}
$$

where the function $P_{[0,1]}(\cdot)$ is applied elementwise to the input matrix, and it has

$$P_{[0,1]}(x) = \begin{cases} x & \text{if } x \in [0,1] \\ 0 & \text{if } x < 0 \\ 1 & \text{if } x > 1. \end{cases}$$

From the projection function (36), we notice that the sparsity of matrix $\mathbf{M}$ from each of projected-gradient descent iterations is monotonously non-increasing. Therefore, we set the stopping criteria to be when $\|\mathbf{M}\|_0$ is close to $n^2 - 2m$. Then, we interpret the resulting matrix as a probabilistic indicator and select the $m$ edge perturbations corresponding to its largest entries to construct the final adversarial perturbation. We summarize this projected gradient descent edge attack in Algorithm 1. Note that gradient-based methods are generally favored for convex objective functions, which does not necessarily apply for all graph filters. However, in the case of adversarial attacks involving a small set of edge perturbations, employing a gradient-based method to find a local optimum can still produce effective adversarial edge perturbations.

---

**Algorithm 1:** `Prob-PGD`

    **Input** : graph $\mathcal{G}$, second-moment matrix $\mathbf{K}$, edge perturbation budget $m$, maximum number
              of iterations $N$, learning rate $\alpha$, tolerance $\tau$
    **Output** : perturbed graph $\mathcal{G}_p$
1   Initialize $\mathbf{M}^{(0)} = 0$;
2   **for** $t = 1 : N$ **do**
3      Update $\mathbf{M}^{(t)}$ via gradient ascent:
4         $\mathbf{M}^{(t)} = f(\mathbf{M}^{(t-1)}) + \alpha \nabla f(\mathbf{M}^{(t-1)})$;
5      Project $\mathbf{M}^{(t)}$ back to the feasible set:
6         $\mathbf{M}^{(t)} \leftarrow \mathbf{\Pi}(\mathbf{M}^{(t)})$     (see Eq. (36));
7      **if** $\|\mathbf{M}^{(t)}\|_0 \leq n^2 - 2m + 2\tau$ **then**
8         **break**
9      **end**
10   **end**
11   Obtain the set of edge perturbations $\mathcal{P}$ by selecting the $m$ pairs $\{u, v\}$ corresponding to the
       largest entries in $\mathbf{M}^{(T)}$;
12   Apply the perturbations in $\mathcal{P}$ to the graph to obtain the perturbed graph $\mathcal{G}_p$;

---

### C.2   Experiment setup

**Graph datasets.** To quantify the effectiveness of adversarial edge perturbation using the two algorithms built on our probabilistic framework, we conduct experiments on a variety of graphs with different structural properties from both synthetic and real-world datasets. Specifically, we consider the following graph datasets:

(1) Stochastic block Models. The stochastic block models (SBM) are random graph models that can generate graphs with *community* structure due to their community-dependent edge probability assignment [19]. In our test experiments, we generate a graph with 40 vertices using SBM with two equally sized communities. The intracommunity edge probability is set to 0.4, reflecting dense connectivity within communities, while the intercommunity edge probability is set to 0.05, indicating sparse connections between communities.

(2) Barabasi-Albert model. The Barabasi-Albert (BA) models generate random *scale-free* graphs through a preferential attachment mechanism, where vertices with higher degree are more likely to be connected by new vertices added to the graph. In our experiments, we sample a graph of 50 vertices, following the preferential attachment scheme, where each new vertex connects to 3 existing vertices.

(3) Watts-Strogatz model. The Watts-Strogatz (WS) model is a random graph model that produces *small-world* properties, i.e., high clustering coefficient and low average path length. This model is built on a underlying ring lattice graph of degree $k$, which exhibits high clustering coefficient. Starting from the ring lattice, each edge is randomly rewired with probability $\beta$ leading to a decrease

in the average path length. Here, the model parameter $\beta$ controls the interpolation between a regular ring lattice ($\beta = 0$) and an Erdos-Renyi random graph ($\beta = 1$). In our experiments, we sample a graph of size 50 from the WS model using $k = 4$ and $\beta = 0.2$.

(4) Random Sensor Network. The random sensor network is a *random geometric graph model*. Here, the vertices are randomly distributed on an underlying space, and the edge probability depends on the geographic distance. We sample a graph with 50 vertices drawn uniformly at random from a 2D coordinate system, with the communication radius to be 0.4.

(5) Zachary's karate club network. Zachary's karate club is a social network of a university karate club [56]. This network captures the interaction of the 34 members of a karate club, and is a popular example of graph with community structure.

(6) ENZYMES. The ENZYMES dataset consists of protein tertiary structures obtained from the BRENDA enzyme database [41]. In this dataset, secondary structure elements (SSEs) of proteins are represented as vertices, and their interactions are captured as edges. Along with the graph structure, this dataset also includes physical and chemical measurements on the SSEs, further providing 18-dimensional numerical graph signals (node features). This dataset contains 600 graphs of the protein tertiary structures. For our test experiment, we randomly select one protein graph from the dataset, which contains 37 vertices, along with the associated 18-dimensional graph signals.

(7) MUTAG. MUTAG is another widely used dataset for graph classification tasks. It consists of 188 graphs, representing mutagenic aromatic and heteroaromatic nitro compounds. Each vertex in the graphs is associated with 7 node features, representing one-hot encoded atom types.

(8) Cora. The Cora dataset contains 2708 scientific publications (vertices) categorized into 7 classes. These documents are connected by 5429 citation links (edges), forming a citation network. Each document is described by a 1433-dimensional sparse bag-of-words feature vector, where each dimension corresponds to a unique word from the corpus vocabulary.

**Graph signal generation.** Graph signals are data attributed to the vertices of a graph, which could be independent of or correlated with the graph structure. Below, we present several popular generative models for graph signals that depend on the underlying graph semantics.

(1) Contextual-SBM: The Contextual stochastic block models (cSBM) are probabilistic models of community-structured graphs with high-dimensional covariate data that share the same latent cluster structure. Let $v \in \{1, -1\}$ be the vector encoding the community memberships. Then a graph signal, condition on $v$ and a latent value $u$, are generated as follows:

$$X_{i,:} = \sqrt{\frac{\mu}{n}} v_i u + Z_i, \tag{37}$$

where $Z_i$ has independent standard normal entries. Therefore, the second-moment matrix $K$ has

$$\mathbf{K}_{ij} = \mathbb{E}[X_{i,:} X_{j,:}^T] = \begin{cases} \frac{\mu}{n} u^2 + 1 \\ -\frac{\mu}{n} u^2 + 1 \end{cases}$$

Intuitively, vertices from the same community have slightly positively correlated covariates, while vertices from different clusters have negatively correlated graph signals.

(2) Smooth graph signal: Smooth graph signals refer to cases where signal values of adjacent vertices do not vary much, a property observed in many real-world datasets, such as geographic datasets. Dong et al. [11] demonstrates that such graph signals can be modeled as samples from a multivariant Gaussian distribution $X \sim \mathcal{N}(\mu, \mathbf{L}^+ + \sigma_\epsilon^2 \mathbf{I})$ where $\mathbf{L}^\dagger$ denotes the Moore-Penrose pseudoinverse of $\mathbf{L}$, and $\sigma_\epsilon \in \mathbb{R}$ represent the noise level. For smooth graph signals, the second-moment matrix has

$$\mathbf{K}_{ij} = \begin{cases} \mu^2 + (\mathbf{L}^+)_{ii} + \sigma_\epsilon^2 & \text{if } i = j \\ \mu^2 + (\mathbf{L}^+)_{ij} & \text{if } i \neq j \end{cases}$$

**Training GCNN models.** For graph classification tasks using the ENZYMES and MUTAG datasets, we partition the dataset into training and test sets using an 80%, 20% split. We pre-train graph

convolutional neural networks (GCNNs), including two-layer GCNs [27], two-layer SGCs [52], and two-layer GIN, where GIN refers to a specialized variant of the original model in which the MLPs are replaced with single-layer perceptrons. All models use a hidden dimension of {32,64}, followed by a ReLU activation, and employ max-pooling as the final readout layer. Training on ENZYMES and MUTAG both use the standard cross-entropy loss. Models are optimized using the Adam optimizer with learning rates between $0.005$ and $0.01$, weight decay $10^{-3}$, and 50 to 100 training epochs. In the adversarial attack experiments, the number of edge perturbations is limited to at most $5\%$ of the total possible edge modifications, i.e., $5\%n^2$, where $n$ is the number of nodes in the graph.

For node classification, we pre-train two-layer GCNs [27], two-layer SGCs [52], and two-layer SUM-GCNs on the original, unperturbed graph, using 140 labeled nodes for training and 1,000 nodes for testing, following the standard setup in [27]. Each GCNN has a hidden dimension of 64 and uses ReLU as the activation function. The models are trained using a standard cross-entropy loss over the labeled training nodes. Training was conducted using the Adam optimizer with a learning rate of $0.01$ and weight decay $5 \times 10^{-4}$ for 50 epochs. In the adversarial attack experiments, each algorithm perturbs 1000 edges. We repeat the experiments five times and report the average classification accuracy.

All experiments were conducted on an NVIDIA V100 GPU with 16GB of memory. Pretraining of the GCNNs on the input graphs was completed within a few minutes. Adversarial attacks on the ENZYMES and MUTAG datasets finished within a few minutes, while attacks on the Cora dataset required slightly more time due to the larger graph size, where `Prob-PGD` and `Wst-PGD` both take nearly 10 minutes to complete, with the maximum number of PGD iterations set to 250.

### C.3  Computing the second-moment matrix K

Our stability analysis so far is based on the knowledge of the second-moment matrix $\mathbf{K}$. In practical scenarios, computing the second-moment matrix is typically straightforward and can be done in several ways. The most straightforward approach is to use the sample second-moment, $\mathbf{K} = \sum_i \mathbf{x}_i \mathbf{x}_i^T$. Beyond simple empirical estimators, more sophisticated methods based on statistical inference are also available, such as adaptive thresholding estimators [6], graphical lasso [15], and linear programming-based techniques [55]. Moreover, domain knowledge can further facilitate the construction of the second-moment matrix. In graph signal processing, meaningful graph structures often align with the distribution of graph signals. For example, in certain sensor networks, geographic proximity tends to produce similar signal patterns, which can be modeled using smoothness assumptions [11].

### C.4  Comparison with the average-case metric for graph filters

With the derived expected embedding stability given from Theorem 1, we now compare our proposed metric on the embedding perturbation with the existing metric based on the worst-case perturbation adopted in [29, 30, 23, 25, 16, 35].

$$\sup_{\|\mathbf{x}\|_2=1} \|g(\mathbf{S})\mathbf{x} - g(\mathbf{S}_p)\mathbf{x}\|_2^2. \tag{38}$$

Recall that in the worst-case metric, the graph signal is normalized to unit length. For a fair comparison, we present in Corollary 4 the expected embedding perturbation when graph signals are sampled from the unit sphere in $\mathbb{R}^n$.

**Corollary 4.** *Consider a graph signal $X$ sampled from the unit sphere in $\mathbb{R}^n$. Then the second-moment matrix $\mathbf{K}$ is positive semi-definite with $\mathrm{Tr}(\mathbf{K}) = 1$. Furthermore, the expected embedding perturbation satisfies*

$$\mathbb{E}_{X \sim \mathcal{D}}[\|\mathbf{E}_g X\|_2^2] = \langle \mathbf{K}, \mathbf{E}_g^T \mathbf{E}_g \rangle \leq \|\mathbf{E}_g\|^2. \tag{39}$$

*Proof.* Note that for any vector $\mathbf{x} \in \mathbb{R}^n$, we have

$$\mathbf{x}^T \mathbf{K} \mathbf{x} = \mathbf{x}^T \mathbb{E}[X X^T] \mathbf{x} = \mathbb{E}[\|\mathbf{x}^T X\|_2^2] \geq 0.$$

Therefore, $\mathbf{K}$ is a positive semidefinite matrix with eigenvalues $\eta_i \geq 0$ for all $i = 1, \ldots, n$. When $X$ is restricted to be sampled only from the unit length sphere, we have that with probability 1,

$$\sum_{i=1}^n X_i^2 = 1.$$

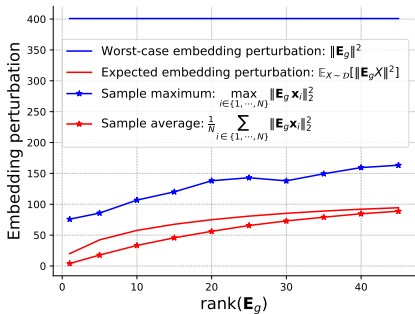

Figure 7: Comparison between the worst-case analysis and average-case analysis. We create an arbitrary filter perturbation $\mathbf{E}_g \in \mathbb{R}^{n \times n}$ with $n = 100$ by sampling i.i.d. entries from $\mathcal{N}(0, 1)$. We further compute different rank approximations on $\mathbf{E}_g$ and test the embedding perturbation induced by each approximation separately. For each scenario, we randomly sample $N = 10000$ graph signals from unit sphere. The red line represents the expected embedding perturbation as defined in (42), with the sample average depicted by the red star-line. Similarly, the blue line represents the worst-case embedding perturbation as defined in (41), with the sample worst-case shown by the blue star-line.

With this, we further have the trace of $\mathbf{K}$

$$\mathrm{Tr}(\mathbf{K}) = \sum_{i=1}^{n} \mathbb{E}[X_i^2] = \mathbb{E}\left[ \sum_{i=1}^{n} X_i^2 \right] = 1. \tag{40}$$

For a given filter perturbation $\mathbf{E}_g$, $\mathbf{E}_g^T \mathbf{E}_g$ is also a positive semidefinite matrix. Therefore, using (10) from Lemma 1, we have

$$\mathbb{E}[\|\mathbf{E}_g X\|_2^2] = \langle \mathbf{E}_g^T \mathbf{E}_g, \mathbf{K} \rangle \leq \|\mathbf{E}_g^T \mathbf{E}_g\| \mathrm{Tr}(\mathbf{K}) = \|\mathbf{E}_g\|^2.$$

$$\square$$

From Corollary 4, it is trivially true that the worst-case embedding perturbation is always an upper bound on the expected embedding perturbation. To further assess how large the gap is between the average-case and worst-case metric, we consider the following three cases, arranged from smallest to largest gap:

**Case 1**. We consider a deterministic input graph signal that achieves the worst possible embedding perturbation. We use $\mathbf{x}_1$ to denote the worst-case signal, which is the eigenvector associated with the largest eigenvalue of $\mathbf{E}_g$. Therefore, the second-moment simply follows as $\mathbf{K} = \mathbf{x}_1 \mathbf{x}_1^T$. Further using the expression of expected embedding perturbation (4), we derive that the average case perturbation is exactly $\|\mathbf{E}_g\|^2$, which matches with the worst-case metric.

**Case 2**. We consider the case where we have no prior information about the graph signal $X$ and simply assume that it is sampled uniformly at random from the unit sphere $\mathbb{S}^{n-1}$. Then we have that the second-moment matrix $\mathbf{K} = \frac{1}{n}\mathbf{I}$ [3].

- If we assess the embedding stability from the worst-case view in (38), then we have

$$\sup_{\|\mathbf{x}\|_2=1} \|g(\mathbf{S})\mathbf{x} - g(\mathbf{S}_p)\mathbf{x}\|_2^2 = \|\mathbf{E}_g\|^2. \tag{41}$$

- If we assess the embedding stability from the average case view, from our analysis in Theorem 1, we have

$$\mathbb{E}_{X \sim \mathrm{Unif}(\mathbb{S}^{n-1})}[\|\mathbf{E}_g X\|_2^2] = \frac{1}{n}\|\mathbf{E}_g\|_F^2. \tag{42}$$

Note that for an arbitrary matrix $\mathbf{M}$ of rank $r$, it always holds that $\|\mathbf{M}\|^2 \leq r^{-1}\|\mathbf{M}\|_F^2$ Therefore, direct comparison between the two bounds (41) and (42) indicates a significant gap between the average-case analysis and worst-case analysis, especially when the change in the graph filter $\mathbf{E}_g$

exhibits a low-rank structure or has large eigengaps. We validate this intuition through numerical simulations on synthetic data, and report the comparison results in Figure 7. From the experiments, we observe that as the rank of the filter perturbation $\mathbf{E}_g$ decreases, the expected embedding perturbation decreases significantly, while the worst-case embedding perturbation remains unchanged. This highlights that for low-rank filter perturbations, the worst-case bound fails to adequately capture the embedding perturbation across a broad range of graph signals.

**Case 3**. If the distribution of $X$ is such that the column vectors of $\mathbf{K}$ are drawn from the null space of $\mathbf{E}_g$, then we have $\mathbb{E}[\|\mathbf{E}_g X\|_2^2] = 0$ with probability 1, and the gap between worst-case and average-case metric is maximized.

The comparison between our probabilistic stability metric and the worst-case metric identifies scenarios where they align and where they differ the most. Other than these special cases, the discrepancy between the average-case metric and worst-case metric is mainly influenced by the rank and eigengap of $\mathbf{E}_g$. Notably, certain graph filters tend to exhibit low rank in $\mathbf{E}_g$, such as graph filters that are linear in the graph shift operator $\mathbf{S}$. For those filters, when only a few edges are perturbed, the resulting $\mathbf{E}_g$ remains sparse, implying its low-rankness. In applications involving these filters, incorporating the average-case stability metric is practically meaningful, as the worst-case metric is proved to be overly conservative and fails to accurately capture the overall embedding perturbations.

### C.5 Embedding perturbation of multilayer GCNNs

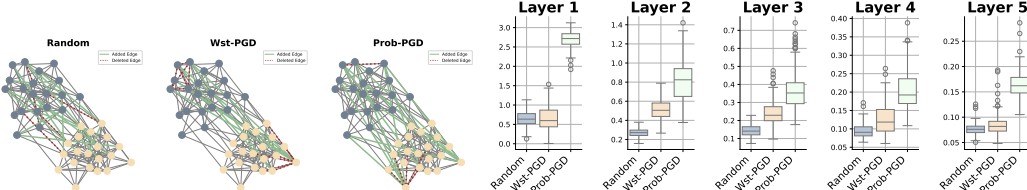

Figure 8: Left side figures are edge perturbations visualization of different algorithms. Box plots report embedding perturbations at different depths of GCN with sigmoid in each layer.

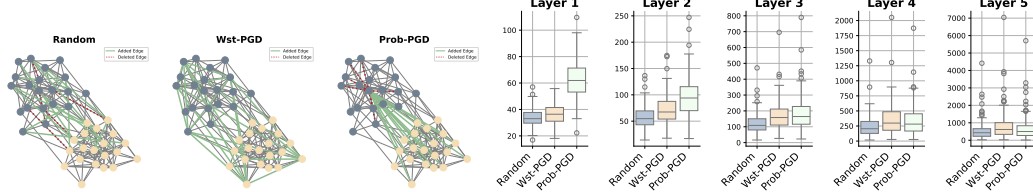

Figure 9: Left side figures are edge perturbations visualization of different algorithms. Box plots report embedding perturbations at different depths of GIN with ReLU in each layer.

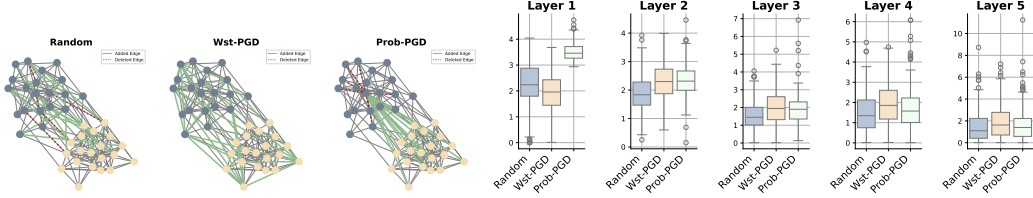

Figure 10: Left side figures are edge perturbations visualization of different algorithms. Box plots report embedding perturbations at different depths of GIN with sigmoid in each layer.

### C.6 Experiments on message passing neural networks

This study mainly focuses on analyzing the stability of Graph Convolutional Neural Networks (GCNNs), which represent one of the two main paradigms in graph neural networks. The other major

paradigm consists of message passing neural networks (MPNNs). While both frameworks aggregate node information based on the underlying graph, MPNNs do not necessarily rely on graph filter convolutions. One notable example of MPNN is the Graph Isomorphism Network (GIN) [53]. In GIN, vertex embeddings are computed recursively. For a given vertex $v$, its embedding at layer $l$ is obtained by aggregating information from its neighbors as well as from itself at the previous layer, according to the following:

$$\mathbf{x}_v^{(l)} = \text{MLP}^{(l)}\left(\left(1 + \epsilon^{(l)}\right) \cdot \mathbf{x}_v^{(l-1)} + \sum_{u \in \mathcal{N}(v)} \mathbf{x}_u^{(l-1)}\right)$$

where $\mathcal{N}(v)$ denotes the set of neighbors of vertex $v$ and $\epsilon^{(l)}$ is either a learnable parameters or a fixed scalar. When the multi-layer perceptron (MLP) is reduced to a single-layer perceptron, the model simplifies to a GCNN with a graph filter of the form $\mathbf{A} + \epsilon^{(l)}\mathbf{I}$. Empirical studies on the stability of GIN with a single-layer perceptron are presented in Section 5. Here, we extend this investigation to GIN architectures with a two-layer perceptron (GIN-2) and a three-layer perceptron(GIN-3), both implemented within a two-layer GIN framework. Using the same experimental settings as in Section 5, we pre-train 10 GINs on unperturbed graphs. Table 5 reports the prediction accuracy and embedding perturbation under various edge perturbation algorithms. Across different datasets and models, `Prob-PGD` induces the largest embedding perturbations, and further results in greater performance degradation for ENZYMES and Cora classification. Overall, GIN models with deeper MLPs exhibit a similar response to edge perturbations as their single-layer counterparts. This observation suggests that our filter-based analysis might be extended to study the stability of MPNNs by investigating graph filters that capture their underlying message passing mechanism.

Table 5: Prediction accuracy of GINs under edge perturbations produced by different methods.

| Dataset | Method | Prediction Accuracy(%) | | Embedding Pert. $\|\cdot\|_F$ | |
| --- | --- | --- | --- | --- | --- |
| | | GIN-2 | GIN-3 | GIN-2 | GIN-3 |
| MUTAG | Unpert. | $82.89_{\pm4.60}$ | $82.11_{\pm5.49}$ | - | - |
| | Random | $70.53_{\pm4.82}$ | $\mathbf{70.53}_{\pm5.37}$ | $28.62_{\pm2.34}$ | $28.43_{\pm3.79}$ |
| | Wst-PGD | $76.32_{\pm8.15}$ | $76.32_{\pm9.04}$ | $37.47_{\pm9.47}$ | $37.37_{\pm11.16}$ |
| | Prob-PGD | $76.58_{\pm8.02}$ | $77.11_{\pm7.81}$ | $\mathbf{96.26}_{\pm11.39}$ | $\mathbf{93.47}_{\pm10.32}$ |
| ENZYMES | Unpert. | $58.92_{\pm3.59}$ | $50.08_{\pm5.30}$ | - | - |
| | Random | $48.25_{\pm3.75}$ | $41.83_{\pm4.15}$ | $92.44_{\pm12.56}$ | $28.35_{\pm5.16}$ |
| | Wst-PGD | $39.33_{\pm4.83}$ | $41.00_{\pm3.14}$ | $295.62_{\pm35.02}$ | $74.06_{\pm12.83}$ |
| | Prob-PGD | $\mathbf{37.00}_{\pm3.25}$ | $\mathbf{35.58}_{\pm3.27}$ | $\mathbf{345.54}_{\pm49.35}$ | $\mathbf{83.59}_{\pm20.24}$ |
| Cora | Unpert. | $73.76_{\pm1.95}$ | $70.72_{\pm2.30}$ | - | - |
| | Random | $69.73_{\pm2.63}$ | $66.95_{\pm3.16}$ | $1129_{\pm344}$ | $1498_{\pm1127}$ |
| | Wst-PGD | $72.55_{\pm2.25}$ | $69.19_{\pm2.20}$ | $16529_{\pm7638}$ | $11105_{\pm6508}$ |
| | Prob-PGD | $\mathbf{57.95}_{\pm3.35}$ | $\mathbf{54.09}_{\pm5.03}$ | $\mathbf{40343}_{\pm23605}$ | $\mathbf{25784}_{\pm13428}$ |

## C.7 Additional experiments on heterophilic datasets

We further conduct empirical study on the following two benchmark heterophilic datasets.

**Chameleon.** The Chameleon dataset contains 2277 nodes representing Wikipedia articles, where edges denote hyperlinks between them. Each node is assigned one of five classes according to the average monthly traffic of the corresponding page. With a homophily score of 0.23, reflecting strong heterophilic properties. Following the standard split, we use 1092 nodes for training, 729 for validation, and 456 for testing.

**Squirrel.** The Squirrel dataset contains 5201 nodes, each representing a Wikipedia article, with edges indicating hyperlinks between pages. Nodes are categorized into five classes based on the average monthly traffic. The dataset has a homophily score of 0.22, reflecting strong heterophilic properties. We follow the standard data split, using 2496 nodes for training, 1664 for validation, and 1041 for testing.

With low homophily scores, these two datasets are challenging for traditional GNNs. Therefore, we adopt the H2GCN model [58], a graph convolutional network specifically designed for heterophilic

Table 6: Results on heterophilic datasets with H2GCN.

| Dataset | Method | Prediction Accuracy (%) | Embedding Pert. $\|\cdot\|_F$ |
|---|---|---|---|
| **Chameleon** | Unperturbed | $55.92 \pm 1.47$ | – |
| | Random | $54.23 \pm 1.16$ | $136.65 \pm 4.27$ |
| | Wst-PGD | $54.87 \pm 1.51$ | $3099.12 \pm 0.00$ |
| | Prob-PGD | $\mathbf{52.96 \pm 1.38}$ | $\mathbf{6392.17 \pm 0.00}$ |
| **Squirrel** | Unperturbed | $50.38 \pm 0.39$ | – |
| | Random | $\mathbf{47.91 \pm 0.47}$ | $578.92 \pm 3.86$ |
| | Wst-PGD | $48.48 \pm 0.73$ | $1127.21 \pm 0.00$ |
| | Prob-PGD | $48.28 \pm 0.73$ | $\mathbf{3908.26 \pm 0.00}$ |

Table 7: Scalability experiments on large-scale graphs.

| Dataset | Nodes | Edges | Embedding Pert. $\|\cdot\|_F$ | | | Memory (MB) | Time (s) |
|---|---|---|---|---|---|---|---|
| | | | Random | Wst | Prob (Ours) | | |
| ogbn-arxiv | 169,343 | 1,166,243 | 4.10 | 14.96 | **25.29** | 390.6 | 537.7 |
| ogbn-mag | 1,939,743 | 21,111,007 | 2.59 | 8.53 | **15.75** | 7362.4 | 1603.0 |
| ogbn-products | 2,449,029 | 61,859,140 | 17.65 | 24.39 | **193.84** | 5833.3 | 2798.5 |
| pokec | 1,632,803 | 30,622,564 | 292.32 | 303.30 | **334.20** | 18040.3 | 1172.1 |

graphs. Given a node feature matrix $\mathbf{X}$ and graph adjacency matrix $\mathbf{A}$, H2GCN constructs node embedding as

$$\mathbf{H} = [\mathbf{X}; \mathbf{AX}; \mathbf{A}^2\mathbf{X}],$$

where the three components correspond to the original node features $\mathbf{X}$, the aggregated 1-hop features $\mathbf{AX}$, and the aggregated 2-hop features $\mathbf{A}^2\mathbf{X}$, respectively. The concatenated embedding $\mathbf{H}$ are then passed through a multi-layer perceptron (MLP) for node classification. For each dataset, we pre-train 10 H2GCNs and report in Table 6 the mean and standard deviation of both prediction accuracy and embedding perturbation under different attack methods by perturbing 1000 edges.

Across both heterophilic graphs, the `Prob-PGD` attack consistently induces the largest embedding perturbation compared to baselines. On the Squirrel dataset, despite producing the most substantial embedding disruption, `Prob-PGD` does not lead to the largest degradation in prediction accuracy. This outcome is not unexpected: `Prob-PGD` is a task-agnostic attack that seeks maximal perturbation in node embeddings rather than direct misclassification. Consequently, the relationship between embedding perturbation and classification accuracy is not always linear.

### C.8   Scalability study on large-scale graphs

Our proposed algorithm `Prob-PGD` has demonstrated efficiency through intensive experiments on medium-sized graphs. The underlying framework, however, is general and can be adapted to large-scale graphs with millions of nodes. In particular, our attack formulation (8) is not restricted to PGD-based optimization and can naturally incorporate alternative search strategies such as greedy heuristics. These approaches iteratively perturb the edge that induces the largest local embedding change and can operate on local subgraphs or within restricted edge subsets. By narrowing the perturbation space, such localized search substantially reduces computational overhead while maintaining competitive attack strength, providing a practical balance between efficiency and optimality.

To provide a concrete example, we adopt a scalable greedy edge-deletion approximation of the original optimization in (8). Specifically, we restrict the perturbation search to existing edges, randomly sample a subset of candidate edges at each iteration, and greedily delete the one that maximizes the attack objective in (8). This stochastic greedy strategy leverages graph sparsity for efficient computation. We denote our approximation by `Prob` and apply the same approximation to the worst-case baseline, denoted as `Wst`. We conduct experiments using the adjacency graph filter on four large-scale OGB benchmarks [20]: ogbn-arxiv, ogbn-mag, ogbn-products, and pokec, where ogbn-mag is heterogeneous and pokec exhibits label heterophily. For each dataset we perform adversarial edge-deletion attacks that remove 200 existing edges. We report average embedding perturbation, peak memory usage, and running time in Table 7.

