# OpenReview forum: "On the Stability of Graph Convolutional Neural Networks: A Probabilistic Perspective"
_NeurIPS.cc/2025/Conference — NeurIPS 2025 poster_

### Official Review · Reviewer_f7yR · 2025-06-24

**Clarity:** 4
**Significance:** 3
**Originality:** 3
**Rating:** 5
**Confidence:** 3

**Summary:**

The paper introduces a probabilistic framework for analyzing the embedding stability of Graph Convolutional Neural Networks (GCNNs). Unlike prior work, which focuses on worst-case perturbations, this paper shifts the focus to expected embedding perturbations. The authors use this formalism to assess the interplay between perturbations, node features and graph topology. They then use this insight to propose a gradient-based perturbation attack to maximally effect the GCNN output, and validate on a range of benchmarks.

**Questions:**

my questions are written within the weaknesses section. I mainly want to understand why some results aren't great and whether stronger baselines or comparisons make sense. I am also interested in getting a better intuition for the size of the bound in Theorem 2.

**Ethical Concerns:**

["NO or VERY MINOR ethics concerns only"]

**Final Justification:**

my accept remains after reading the rebuttal

**Limitations:**

yes

**Quality:**

3

**Strengths And Weaknesses:**

**Strengths**

- The limitations of prior works in terms of focusing on worst-case perturbations is clearly outlined and addressed with their probabilistic approach. This limitation is problematic to the community so an attempt to address this is impactful. The study on the contextual stochastic block model (cSBM) offers both theoretical and empirical justification for the heuristic perturbation strategies observed in prior work.

- The theoretical contributions of the paper are very compelling. The theoretical contributions alone are a substantial contribution. For example the exact characterization of the stability of graph filters and the analysis and connections between perturbations, node features and graph topology.

**Weaknesses**

- Some of the results are inconclusive about the benefits of the approach. For example, Prob-PGD causes a larger degradation on MUTAG than random with a GCN but not with a GIN. Is there any reason for this? Given that results like this highlight the strong dependance on the model architecture, the generalization of the approach may not be strong and thus be less impactful. It might be worth at least showing that your perturbation perturbs the embedding more than a random perturbation for MUTAG with GIN. A similar analysis has also been performed in [1] and would at least show that the reason for this result is some complexity in the label space/training rather than the attacks ability to maximize embedding distance after perturbation.

- As we increase the number of layers, the bound in Theorem 2 gets worse with these recursive constants due to dependencies and compounding nonlinearities. It may be useful to have some toy experiments with real-datasets (eg. MUTAG) to get a sense of the size of these constants. How bad is this bound with increasing layers?

- Papers like [1] and [2] have discussed pseudometrics between graphs. [1] provides an upper-bound on the embedding distance with respect to edge deletions. I understand your method can provide a probabilistic description of the stability unlike these approaches. But I don't necessarily see why an attack based on your measure would be better. Additionally, you could also use standard structural measures to do the attacks (eg. Edge Betweenness Centrality.) Perhaps comparing to stronger baselines may be useful for assessing the benefits of the approach.

[1] Chuang et al. Tree Mover’s Distance: Bridging Graph Metrics and Stability of Graph Neural Networks. NeurIPS 2022.

[2] Rauchwerger et al. Generalization, Expressivity, and Universality of Graph Neural Networks on Attributed Graphs. ICLR 2025.

---

> ### Author Rebuttal · Authors · 2025-07-30
>
> > (1) Some of the results are inconclusive about the benefits of the approach. For example, Prob-PGD causes a larger degradation on MUTAG than random with a GCN but not with a GIN. Is there any reason for this?
>
> We thank the reviewer for the thoughtful observation. To address this concern, we now report in Table 1 **both classification accuracy and embedding perturbation magnitude ** (measured by the Frobenius norm), averaged over 10 independently trained models. This enables a clearer comparison of the attack strength across different architectures.
>
> | Dataset  | Method   | GCN Acc. (%)       | GIN Acc. (%)       | SGC Acc. (%)       | GCN ebd. ($\|\cdot\|_F$) |  GIN  ebd ($\|\cdot\|_F$)|  SGC ebd  ($\|\cdot\|_F$)|
> |----------|----------|--------------------|---------------------|---------------------|------------------|-------------------|-------------------|
> | **MUTAG** | Unpert.  | 70.00 ± 2.68        | 82.89 ± 4.60         | 68.95 ± 1.58         | -                | -                 | -                 |
> |          | Random   | 73.16 ± 1.05        | **70.53 ± 4.82**         | 72.89 ± 1.21         | 2.24 ± 0.15      | 28.62 ± 2.34       | 3.57 ± 0.17        |
> |          | Wst-PGD  | 69.21 ± 2.37        | 76.32 ± 8.15         | 68.16 ± 1.42         | 1.95 ± 0.13      | 37.47 ± 9.47       | 3.11 ± 0.16        |
> |          | Prob-PGD | **42.89 ± 4.86**    | 76.58 ± 8.02         | **41.58 ± 3.87**     | **3.09 ± 0.16**  | **96.26 ± 11.39**  | **5.21 ± 0.25**    |
> | **ENZYMES** | Unpert.  | 56.33 ± 3.10        | 56.75 ± 2.99         | 54.75 ± 3.48         | -                | -                 | -                 |
> |          | Random   | 49.58 ± 3.93        | 45.83 ± 2.64         | 45.50 ± 2.94         | 15.3 ± 0.7       | 81.3 ± 11.3        | 36.4 ± 1.2         |
> |          | Wst-PGD  | 52.58 ± 3.56        | 39.75 ± 3.29         | 48.75 ± 4.27         | 11.5 ± 0.5       | 264.9 ± 34.7       | 27.2 ± 0.9         |
> |          | Prob-PGD | **44.25 ± 2.09**    | **36.75 ± 4.62**     | **40.75 ± 3.83**     | **17.9 ± 0.9**   | **297.9 ± 57.1**   | **43.6 ± 1.6**     |
> | **Cora**   | Unpert.  | 80.06 ± 0.51        | 74.92 ± 1.86         | 80.33 ± 0.29         | -                | -                 | -                 |
> |          | Random   | 78.30 ± 0.82        | 71.75 ± 1.85         | 78.12 ± 0.64         | 93 ± 4           | 974 ± 496          | 136 ± 4            |
> |          | Wst-PGD  | 78.47 ± 0.44        | 73.76 ± 2.07         | 78.30 ± 0.33         | 97 ± 7           | 16901 ± 12495      | 151 ± 5            |
> |          | Prob-PGD | **77.88 ± 0.33**    | **56.67 ± 4.57**     | **77.49 ± 0.29**     | **106 ± 5**      | **43565 ± 33035**  | **160 ± 6**        |
>
> **Table 1: Prediction accuracy of GCNNs under edge perturbations produced by different methods.**
>
>
> We observe that across all architectures and datasets, including MUTAG with GIN, our proposed Prob-PGD **consistently produces the largest embedding perturbations**, which aligns with our theoretical motivation. This confirms that Prob-PGD **effectively maximizes structural sensitivity at the representation level, independent of the model used**.
> We note that the final **accuracy drop also depends on the task-specific decision boundary**. In particular, for GIN on MUTAG, despite Prob-PGD producing the most substantial embedding shift, the prediction degradation is less significant. This suggests that the perturbed representations remain on the same side of the classifier's (MLP’s) decision boundary, which is not unexpected for a task-agnostic embedding attack. We emphasize that a large embedding perturbation is a necessary (but not sufficient) condition for adversarial vulnerability across diverse downstream tasks, and thus offers **a general-purpose tool for assessing the model robustness**. We will include this clarification in the revision.
>
> > (2) comparison to [1,2]?
>
> We acknowledge that alternative metrics such as those in [1,2] offer valuable insights and have their own advantages. Beyond the interpretability enabled by our probabilistic formulation, our proposed metric provides several key benefits:
>
> (a) Generality: Our metric is applicable to GCNNs with a broad range of graph filter structures, such as low-pass, high-pass, and heat diffusion filters, that could involve higher-order structures. In comparison to [1], where the analysis is tightly tailored to  GIN-style message passing, our approach provides insights into spectral GNN settings.
>
> (b) tightness for single-layer models: Most prior metrics provide only upper bounds characterization on the model stability, often due to mathematical intractability or by definition. Optimizing these upper bounds does not guarantee a strong attack. In contrast, for many single-layer GCNNs, our study yields a tight, closed-form characterization of expected embedding perturbation, which is more direct for designing effective attack strategies.
>
> We will add a more detailed comparison with these approaches in the revised manuscript’s related work section to highlight these distinctions.
>
> > (3) Comparing to stronger baselines?
>
> We thank the reviewer for the constructive suggestion. In response, we have included preliminary experiments incorporating centrality-based attacks as an additional baseline.  Specifically, we compare the classification accuracy and embedding norm perturbation under various attacks on the ENZYMES dataset using SGC and GIN models:
>
> **ENZYMES dataset using SGC model.**
>
> | Method      | Accuracy (%)       | Embedding pert. ($\|\cdot\|_F$) |
> |-------------|--------------------|-------------------------------|
> | Test        | 53.75 ± 3.44       | -                             |
> | Random      | 47.58 ± 3.38       | 38.14 ± 1.29                  |
> | Centrality  | **34.17 ± 2.58**       | **85.63 ± 2.70**                  |
> | Wst-PGD     | 48.42 ± 2.90       | 28.13 ± 0.98                  |
> | Prob-PGD    | 43.08 ± 3.59    | 45.72 ± 1.45                 |
>
>
> ---
>
> **ENZYMES dataset using GIN model.**
>
> | Method             | Accuracy (%)           | Embedding pert. $\|\cdot\|_F$ |
> |--------------------|------------------------|--------------------------------------|
> | Unperturbed  | 56.58 ± 4.19           | -                                    |
> | Random             | 45.75 ± 3.90           | 118.12 ± 7.37                        |
> | Centrality    | 41.58 ± 2.43           | 83.00 ± 4.55                         |
> | Wst-PGD            | 38.67 ± 4.64           | 380.03 ± 20.75                       |
> | Prob-PGD           | **37.17 ± 3.86**       | **492.80 ± 42.41**                   |
>
> As shown in the above results, both centrality-based attacks and Prob-PGD produce competing results, leading to significantly larger embedding shift and accuracy drop than other baselines. We also note an important distinction: centrality-based attacks are distribution-independent, and thus their effectiveness can vary with changes in the underlying feature distribution. We will include more empirical studies with stronger baselines in the revision, together with a more detailed analysis to better interpret the experimental results.
>
> > (4) Tightness of the bound in Theorem 2?
>
> We thank the reviewer for this constructive suggestion. While we did not include such an empirical analysis in the current response due to time constraints, we agree that it is an important direction to understand the behavior of the bound with increasing depth. We plan to include empirical results (e.g., similar to Figure 5 from [1]) in the revision or during the author-response discussion phase to illustrate how the bound scales with depth and to provide insight into its tightness.

---

> ### Comment · Reviewer_f7yR · 2025-08-05
>
> Thank you for your detailed response! I appreciate the extra effort which has gone into improving the paper. The authors have addressed my concerns and I maintain my preference for accepting the paper.

---

### Official Review · Reviewer_MeYi · 2025-06-27

**Clarity:** 3
**Significance:** 2
**Originality:** 3
**Rating:** 4
**Confidence:** 3

**Summary:**

The paper departs from the prevailing worst-case view of GNN robustness and introduces a distribution-aware framework that measures expected embedding perturbations when edges are added or deleted at test time. Closed-form expressions are obtained for graph filters, and depth-dependent upper bounds are derived for multilayer GNNs. The theory yields structural insights that explain which edge sets hurt most and motivates a new projected-gradient attack, Prob-PGD, that maximises expected embedding change. Experiments on six benchmark graphs and three downstream tasks demonstrate that Prob-PGD produces larger embedding shifts and bigger accuracy drops than strong baselines, confirming the practical value of the analysis.

**Questions:**

NA

**Ethical Concerns:**

["NO or VERY MINOR ethics concerns only"]

**Final Justification:**

I think my score is reasonable.

**Limitations:**

yes

**Quality:**

3

**Strengths And Weaknesses:**

**Strengths**

1. First probabilistic stability framework for GNNs, giving a realistic alternative to worst-case analysis.

2. Clear structural insight: links autocorrelation and edge-intersection patterns to vulnerability, rationalising prior heuristic attacks.

3. Prob-PGD, derived directly from theory, consistently outperforms random and worst-case PGD in both embedding and task metrics.

**Weaknesses**

1. Focuses only on evasion (test-time) attacks; poisoning scenarios are not studied.

2. Experiments use small graphs; scalability to million-node graphs is untested.

---

> ### Author Rebuttal · Authors · 2025-07-30
>
> >(1) poisoning attack?
>
> We appreciate the reviewer’s comment. Our focus on evasion attacks is deliberate, as our theoretical framework is built on understanding the inference-time stability, which is directly related to test-time behavior. While poisoning attacks are indeed important, they operate through fundamentally different mechanisms.
>
> To clarify the distinction:
>  - Evasion attacks manipulate the test data while keeping the model parameters fixed.
>  - Poisoning attacks perturb the training data to influence the learned model parameters, which subsequently affects test-time performance.
>
> Consequently, the relevant notions of stability also differ. **Evasion attacks are closely connected to inference-time (test-time) stability**, which is the focus of our study. (For a more detailed explanation of the importance of this topic, we refer the reviewer to our response to Reviewer 6sQf.) In contrast, **poisoning attacks are more closely related to algorithmic stability**, i.e., how sensitive a machine learning algorithm is to small changes in the training data [1]. While there may be conceptual connections between these two stability notions, they pertain to distinct aspects of the learning process.
>
> Given this distinction, our theoretical results on inference-time stability do not directly extend to poisoning scenarios.
>
> Despite it being outside our theoretical framework, we conduct the following experiment to evaluate empirically how Prob-PGD performs under a poisoning attack.
>
> - Set up:
> We use a GCN for graph classification on the MUTAG dataset. In the poisoning attack, we randomly choose 20% of the graphs from the training set and perturb 5% of their edges using different attack algorithms. We train GCN models (with fixed hyperparameters) on these poisoned datasets separately and report the test accuracy averaged over 10 runs.
> - Results :
> | Method       | Accuracy (%) |
> |--------------|--------------------|
> | Unperturbed  | 69.21       ±1.21   |
> | Random       | 66.58        ±1.21   |
> | Wst-PGD      | 70.79         ±2.19   |
> | Prob-PGD     | 69.47         ±2.93   |
>
> These results suggest that the GCN model maintains a certain level of robustness to poisoning attacks, where test accuracy is not significantly degraded across methods. However, we emphasize that **these empirical observations fall outside the scope of our theoretical framework, and interpreting them rigorously would require new theoretical developments**. We thank the reviewer for encouraging this important line of inquiry, and we view it as a promising direction for future research.
>
> [1] Saurabh Verma, Zhi-Li Zhang, Stability and Generalization of Graph Convolutional Neural Networks, ACM KDD'19
>
>
> >(2) scalability to large graphs?
>
> We thank the reviewer for the question regarding scalability. Our algorithm Prob-PGD is efficient on moderate-sized graphs(in practice, ~0.8 seconds per iteration on Cora on an A100GPU).
> Scaling to graphs with millions or billions of nodes could be challenging. The primary bottleneck lies in the $O(n^2)$ **time and memory complexity of computing gradients over the full adjacency matrix**. However, it is important to note that this complexity is not unique to our approach but is a **general limitation of edge attack algorithms that operate over the full edge space**. Since the search space includes all possible edge and non-edge positions, the number of potential perturbations grows quadratically with the number of nodes.
>
> We agree that studying large-scale datasets would be valuable and will consider this in a future extension of our study.  Here, we outline several possible directions to improve scalability:
>
> (a)**mini-batch gradient**: Rather than computing the gradient over the entire graph, we can approximate it using a mini-batch of b nodes per iteration. This reduces both the time and memory complexity per iteration to O(b^2), making it feasible for large graphs.
>
> (b)**greedy&local search**: Our methodology (see line 310) is not restricted to PGD and allows alternative algorithms, such as greedy heuristics that recursively perturb one edge with the highest embedding impact. These methods can be applied to local subgraphs or a subset of edges to narrow the search space and reduce computational overhead.
>
>
> While these approximations trade off some optimality, they offer promising routes for handling large-scale networks.  To demonstrate feasibility, we conducted an experiment using a greedy edge deletion strategy, where the search space is restricted to existing edges, thereby leveraging graph sparsity for efficient computation.
>
>
> - Dataset: ogbn-arxiv (169,343 nodes, 1,166,243 edges, 128-dimensional features)
> - Perturbation: 20 edge deletions
> - Runtime: 9 minutes 53 seconds
> - Memory usage: 1182.85 MB
> - Average embedding perturbation: 2.43 (random baseline produces 1.23)
>
> These results show that scalable approximations of our method can nearly double the magnitude of embedding perturbations compared to random baselines, enabling efficient edge attacks on large graphs. Due to time and computational resource constraints, we were unable to include additional large-scale datasets. We will include a more detailed discussion on scalability in our revised version.

---

> > ### Comment · Reviewer_MeYi · 2025-08-03
> > **Follow-up**
> >
> > Thanks for the responses. Due to the scalability limitation, I believe my current score is reasonable.

---

> > > ### Author Response · Authors · 2025-08-05
> > >
> > > We thank the reviewer for the follow-up and apologize for any confusion caused by our previous response. Our intention was to convey that **our algorithm can be scaled to handle million-node graphs through different approximation strategies**. While our **initial experiment was conducted on a moderately large dataset due to resource constraints**, we did not mean to suggest that our method is inherently limited only to small or medium-sized graphs.
> > > To address this more concretely, we have conducted an additional experiment on a million-node graph:
> > >
> > > Dataset: ogbn-mag
> > > - Nodes: **1,939,743**
> > > - Edges: 21,111,007
> > >
> > > Attack Method: Local greedy edge deletion of 20 edges (as described in our earlier response).
> > > - Peak Memory Usage: 7362.44 MB
> > > - Running Time: 633.64 seconds
> > >
> > > Embedding Perturbation:
> > > - **Greedy (our method): 1.0656**
> > > - Random baseline: 0.7423
> > >
> > > These results demonstrate that **our method can be scaled to million-node graphs with reasonable efficiency and impact**. We hope this clarifies our position and addresses the reviewer’s concern.
> > > Given this new evidence of both feasibility and scalability, we would respectfully ask the reviewer to consider revisiting their score. We remain open to further clarification if any part of our response is still unclear.

---

> > > > ### Author Response · Authors · 2025-08-08
> > > >
> > > > We took the reviewer’s comment on scalability very seriously and apologise that our previous response was not entirely satisfactory. In this revised response, we provide a more comprehensive and detailed follow-up, including:
> > > >
> > > > - A more explicit description of our scalable approximation algorithm
> > > > - Additional experiments on datasets of varying sizes and properties
> > > > - A broader set of baseline comparisons
> > > >
> > > > We already provided general directions of how to scale up our algorithm above, and for the experiments below we adopt an approximation of the algorithm by restricting the attack to deleting existing edges and greedily searching for edges to delete, where in each iteration of the greedy search we randomly sample candidate edges to delete (note that such a strategy is not uncommon in the graph learning literature; for example, both GraphSAGE and DropEdge employed similar strategies, albeit for different motivations). For ease of reference, we continue to refer to the adapted algorithms as Prob-PGD and Wst-PGD, though we note that both methods reported now are approximate versions designed for scalability. We conduct adversarial edge-deleting attacks by removing 200 existing edges, and compare Prob-PGD against random baseline and Wst-PGD (using the same approximation technique for a fair comparison).
> > > >
> > > > We test on four datasets that are considered large-scale in the literature (according to [1] for example): ogbn-arxiv, ogbn-mag, ogbn-products, and pokec. (we skipped ogbn-proteins in [1] due to lack of node features). Additionally, ogbn-mag is an example of a heterogeneous graph (with different node types), while pokec is a graph with heterophily in node labels. Below, we report for each dataset its average embedding perturbation, computed as $1/d \sum_{i=1}^d \\|E_g X_i\\|_2$, where $x_i$ denotes the node feature and $d$ is the feature dimension.
> > > >
> > > > **(a) obgn-arxiv (number of vertices=169,343, number of edges =1,166,243)**
> > > >
> > > > Embedding perturbation:
> > > >
> > > > Random: 4.0979
> > > >
> > > > Wst-PGD: 14.9566
> > > >
> > > > Prob-PGD (Ours): 25.2936
> > > >
> > > > - Peak memory usage (Prob-PGD): 390.58 MB
> > > > - Running time (Prob-PGD): 537.74 seconds
> > > >
> > > > **(b) obgn-mag (number of vertices=1,939,743, number of edges=21,111,007)**
> > > >
> > > > Embedding perturbation:
> > > >
> > > > Random: 2.5874
> > > >
> > > > Wst-PGD: 8.5337
> > > >
> > > > Prob-PGD (Ours): 15.7489
> > > >
> > > > - Peak memory usage (Prob-PGD): 7362.41 MB
> > > >
> > > > - Running time (Prob-PGD): 1603.04 seconds
> > > >
> > > > **(c) obgn-products (number of vertices = 2,449,029, number of edges = 61,859,140)**
> > > >
> > > > Embedding perturbation:
> > > >
> > > > Random: 17.6481
> > > >
> > > > Wst-PGD: 24.3905
> > > >
> > > > Prob-PGD (Ours): 193.8391
> > > >
> > > > - Peak memory usage (Prob-PGD): 5833.25 MB
> > > >
> > > > - Running time (Prob-PGD): 2798.54 seconds
> > > >
> > > > **(d) pokec (number of vertices = 1,632,803, number of edges = 30,622,564)**
> > > >
> > > > Embedding perturbation:
> > > >
> > > > Random: 292.3226
> > > >
> > > > Wst-PGD: 303.3037
> > > >
> > > > Prob-PGD (Ours): 334.1984
> > > >
> > > > - Peak memory usage(Prob-PGD): 18040.25 MB
> > > >
> > > > - Running time(Prob-PGD): 1172.07 seconds
> > > >
> > > > From the results above, we can see that **Prob-PGD consistently outperforms both the random baseline and Wst-PGD in terms of embedding perturbation**. We hope these experiments demonstrate that our algorithm (with a reasonable approximation) can indeed **scale up to graphs with millions of nodes and tens of millions of edges**.
> > > >
> > > > We thank the reviewer again for their engagement and remain at their disposal for any further comments or suggestions they may have.
> > > >
> > > > [1] Classic GNNs are Strong Baselines: Reassessing GNNs for Node Classification, NeurIPS 2024.

---

> > > > > ### Author Response · Authors · 2025-08-09
> > > > > **Thank you for your engagement**
> > > > >
> > > > > Dear reviewer,
> > > > >
> > > > > We would like to express our sincere appreciation for your engagement in the review and rebuttal discussion, which we believe has helped improve the quality of the paper. We will incorporate the corresponding modifications in the final version accordingly.
> > > > >
> > > > > Best regards,
> > > > >
> > > > > The authors

---

### Official Review · Reviewer_6sQf · 2025-06-29

**Clarity:** 2
**Significance:** 3
**Originality:** 3
**Rating:** 5
**Confidence:** 3

**Summary:**

This paper studies how the perturbation on the graph would affect the output embeddings from a probabilistic view.  It studies the relation of the expected embedding perturbation with the input feature distribution and the changes in the graph structure.  The authors also designed a fast algorithm using projected gradient descent to find the adversarial examples with larger embedding pertubations compared to the baseline, as shown by the empirical results.

**Questions:**

Some minor suggestions:

- The legend of Figure 1 can be put outside of the contents.

- In line 164, it should be $S_p$.

- Corollary 3 seems to be incomplete.

- The main contribution part in section 1 can be more concise.

**Ethical Concerns:**

["NO or VERY MINOR ethics concerns only"]

**Final Justification:**

I keep my original score and my concerns are addressed.

**Limitations:**

Yes.

**Paper Formatting Concerns:**

No.

**Quality:**

3

**Strengths And Weaknesses:**

Strength:

1. Comprehensive probabilistic bounds are derived to show the factors contributing to the embedding perturbation.
2. The experiment results show a significantly larger perturbation compared to the baseline.

Weakness:

The motivation is not strongly established.  I wonder what is the real-world application of the research.  In another word, why is this topic important?

---

> ### Author Rebuttal · Authors · 2025-07-30
>
> We thank the reviewer for the helpful suggestions, and we will address them correspondingly in our revised manuscript.  We would like to explain a bit why this topic is important.
>
> While GCNNs have achieved remarkable success across various domains, their deployment in practice faces a critical challenge: the graph structure at inference time may differ from what the model was trained on, due to either noise, dynamics, or adversarial manipulations. Retraining models to adapt to each variation is often impractical due to time, cost, and data constraints. This makes it essential to understand and quantify how robust pre-trained GCNNs are to such perturbations.
>
> The following practical scenarios could further help explain this need:
>
> **Defensive robustness** in social media platforms: Pre-trained graph-based misinformation detectors may be targeted by adversarial agents (e.g., bots) who add or delete edges (friendships, follows) to evade detection. Understanding embedding stability helps assess whether such detectors are reliable under manipulative behaviors.
>
> In **dynamic environments** such as communication networks, traffic systems, or recommender systems, graph structures evolve over time. Our framework enables practitioners to assess whether a model trained on an earlier graph snapshot can still be used effectively as the graph changes, which provides a critical tool for model updating planning.

---

> > ### Comment · Reviewer_6sQf · 2025-08-05
> >
> > Thanks the authors for your reponse.

---

> > > ### Author Response · Authors · 2025-08-09
> > > **Thank you for your engagement**
> > >
> > > Dear reviewer,
> > >
> > > We would like to express our sincere appreciation for your engagement in the review and rebuttal discussion, which we believe has helped improve the quality of the paper. We will incorporate the corresponding modifications in the final version accordingly.
> > >
> > > Best regards,
> > >
> > > The authors

---

### Official Review · Reviewer_BGAr · 2025-07-03

**Clarity:** 3
**Significance:** 3
**Originality:** 3
**Rating:** 4
**Confidence:** 4

**Summary:**

This paper investigates the stability of Graph Convolutional Neural Networks (GCNNs) to small perturbations in the graph structure. The theoretical analysis in this paper is performed from a probabilistic perspective, which accounts for the underlying distribution of graph signals and perturbations. The paper includes comprehensive empirical evaluations, including testing robustness against adversarial attacks, which emphasize the importance of considering data distributions in stability analysis.

**Questions:**

I suggest that the authors also review the recent literature on coVariance neural networks (VNNs), which study data-driven perturbations to covariance graphs in GCNNs and are perhaps more closely related to their work.

[a] Sihag, Saurabh, et al. "coVariance neural networks." Advances in neural information processing systems 35 (2022): 17003-17016.

[b] Cavallo, Andrea, Mohammad Sabbaqi, and Elvin Isufi. "Spatiotemporal covariance neural networks." In Joint European Conference on Machine Learning and Knowledge Discovery in Databases, pp. 18-34. Cham: Springer Nature Switzerland, 2024.

[c] Cavallo, A., Navarro, M., Segarra, S., & Isufi, E. (2025, April). Fair covariance neural networks. In ICASSP 2025-2025 IEEE International Conference on Acoustics, Speech and Signal Processing (ICASSP) (pp. 1-5). IEEE.

**Ethical Concerns:**

["NO or VERY MINOR ethics concerns only"]

**Limitations:**

I did not find an explicit discussion on limitations of this work.

**Paper Formatting Concerns:**

No paper formatting concerns.

**Quality:**

3

**Strengths And Weaknesses:**

**Strengths**

1. The paper provides a probabilistic scenario to stability of GCNNs, which is likely to provide more realistic insights relative to the worst case stability analyses in the literature.
2. Theoretical results on stability are rigorous and cover multi-layer GCNNs.
3. The theory presented in this paper further motivates a distribution-aware attack algorithm Prob-PGD, which produces greater performance degradation relative to other baselines.

**Weaknesses**

No notable weaknesses.

---

> ### Author Rebuttal · Authors · 2025-07-30
>
> We thank the reviewer for bringing this relevant line of work to our attention. CoVariance Neural Networks (VNNs) use the covariance matrix of graph signals (i.e., the covariance graph) as the underlying graph structure for GCNNs.
>
> In the stability analysis of VNN [a,b,c], the authors consider perturbation on the covariance graph  $C-\hat{C}$, which arises due to estimation errors from a finite number of graph signal samples. These perturbations are data-dependent, as they depend on the underlying distribution of graph signals. The corresponding output stability is assessed using a worst-case metric, measured via the spectral (operator) norm $\|H(C)-H(\hat{C})\|$ between the perturbed and original covariance graph filters.
>
> In contrast, our work considers more general GCNNs, and the graph perturbations are also generic. The data-dependence in our stability analysis arises from the notion of stability itself, where we show that our introduced metric, the expected embedding perturbation, depends on the graph signal distribution. We summarize the key differences as follows:
>
>
> | Category         | Stability [a, b, c]               | Ours                            |
> |------------------|-----------------------------------|----------------------------------|
> | Perturbation     | Estimation error                  | Generic edge perturbation       |
> | Stability Metric | Worst-case                        | Average-case                     |
> | Architecture     | Covariance-based GCNNs            | General GCNNs                    |
>
>
>
> Despite these differences, both approaches adopt a distribution-aware perspective and highlight connections between network stability and the second-order statistics of graph signals. This opens the door to interesting directions for combined research. For example, our probabilistic framework could complement the VNN line of research by offering an average-case sensitivity analysis of VNNs, in which both the perturbation model and the notion of stability are tied to the underlying distribution of graph signals. Understanding stability from this perspective could be mathematically interesting and potentially lead to more representative and interpretable measures of robustness for VNNs.
>
> We will include a more detailed discussion and comparison to VNN-related works [a, b, c] in the revised manuscript and highlight the conceptual and methodological connections.

---

### Official Review · Reviewer_SjSK · 2025-07-03

**Clarity:** 4
**Significance:** 2
**Originality:** 3
**Rating:** 3
**Confidence:** 4

**Summary:**

This paper investigates the stability of Graph Convolutional Neural Networks (GCNNs) from a probabilistic perspective. Specifically, it studies the sensitivity of GCNN embeddings to small perturbations in graph topology. Unlike prior research focusing primarily on worst-case perturbations, this paper proposes a novel distribution-aware formulation, analyzing expected embedding perturbations across various input data. The authors derive exact characterizations and upper bounds for embedding perturbations in both graph filters and multilayer GCNNs. They further interpret these theoretical findings through structural insights, clarify conditions under which certain perturbations are more impactful, and propose a novel adversarial attack method (Prob-PGD) validated through extensive experiments.

**Questions:**

(1) How well does the proposed framework scale to very large graphs such as ogbn, Reddit, or Aminer-CS? Have the authors conducted any runtime or memory usage comparisons on such datasets?

(2) How might the framework perform on heterophilous or mixed-type graphs? Would the authors consider including a discussion or additional evaluations on this point?

(3) Considering that adversarial defense against structural attacks is a well-studied area, could the authors expand the empirical comparison to include more recent or stronger baselines [1, 2, 3, 4]?

**Ethical Concerns:**

["NO or VERY MINOR ethics concerns only"]

**Final Justification:**

The authors have made a commendable effort in addressing several concerns, particularly those related to autocorrelation estimation and evaluation on heterophilic graphs. However, the scalability claims of the proposed method remain insufficiently supported. The results on small-scale datasets such as Cora, which contain only thousands of nodes, do not adequately demonstrate the method's practical applicability to real-world graph settings that involve significantly larger and more complex structures. Moreover, the additional experiment on ogbn-arxiv evaluates only a simplified greedy variant of the method and compares it solely against a random baseline, which limits the strength of the empirical evidence. As a result, I intend to maintain my original score.

**Limitations:**

yes

**Paper Formatting Concerns:**

No major formatting issues.

**Quality:**

3

**Strengths And Weaknesses:**

Strengths:

(1) The proposed probabilistic framework extends beyond traditional worst-case analyses, providing deeper insights into embedding stability considering realistic data distributions.

(2) Rigorous derivations of stability measures for graph filters and multilayer GCNNs enhance the theoretical understanding of these models.

(3) The paper is well-structured, clear, and self-contained, with theoretical results clearly presented and complemented by insightful empirical analysis.

Weaknesses:

(1) The computational efficiency and scalability to very large-scale graphs are not extensively addressed, potentially limiting applicability in large-scale graphs, such as ogbn, Reddit, and Aminer-CS.

(2) The framework requires accurate estimation of the signal autocorrelation matrix, which may be challenging in real-world scenarios with limited or noisy data.

(3) The evaluation is limited to homophily graphs. The author should add a discussion on that.

(4) The major concern is with the empirical study in this paper. Adversarial defense against structural adversarial attacks is a widely studied topic [1,2,3,4]. However, only a limited number of works are discussed and compared in the experiments.

[1] Reliable representations make a stronger defender: Unsupervised structure refinement for robust gnn

[2] Self-guided robust graph structure refinement

[3] Adversarial robustness in graph neural networks: A Hamiltonian approach

[4] Elastic graph neural networks

---

> ### Author Rebuttal · Authors · 2025-07-31
>
> We thank the reviewer for their acknowledgement of the strengths of our work, and for raising insightful comments to help us further improve the paper. Below we provide detailed response to each comment:
>
> > (1) scalability to very large-scale graphs?
>
> We thank the reviewer for the question regarding scalability. Our algorithm Prob-PGD is efficient on moderate-sized graphs (in practice, ~0.8 seconds per iteration on Cora on an A100GPU).
> Scaling to graphs with millions or billions of nodes could be challenging. The primary bottleneck lies in the $O(n^2)$ time and memory complexity of computing gradients over the full adjacency matrix. However, it is important to note that this complexity is not unique to our approach but is a **general limitation of edge attack algorithms that operate over the full edge space**. Since the search space includes all possible edge and non-edge positions, the number of potential perturbations grows quadratically with the number of nodes.
>
> We agree that studying large-scale datasets would be valuable and will consider this in a future extension of our study.  Here, we outline several possible directions to improve scalability:
>
> (a) **mini-batch gradient**: Rather than computing the gradient over the entire graph, we can approximate it using a mini-batch of b nodes per iteration. This reduces both the time and memory complexity per iteration to O(b^2), making it feasible for large graphs.
>
> (b) **greedy & local search**: Our methodology (see line 310) is not restricted to PGD and allows alternative algorithms, such as greedy heuristics that recursively perturb one edge with the highest embedding impact. These methods can be applied to local subgraphs or a subset of edges to narrow the search space and reduce computational overhead.
>
>
> While these approximations trade off some optimality, they offer promising routes for handling large-scale networks.  To demonstrate feasibility, we conducted an experiment using a greedy edge deletion strategy, where the search space is restricted to existing edges, thereby leveraging graph sparsity for efficient computation.
>
>
> - Dataset: ogbn-arxiv (169,343 nodes, 1,166,243 edges, 128-dimensional features)
> - Perturbation: 20 edge deletions
> - Runtime: 9 minutes 53 seconds
> - Memory usage: 1182.85 MB
> - Average embedding perturbation: 2.43 (random baseline produces 1.23)
>
> These results show that scalable approximations of our method can nearly double the magnitude of embedding perturbations compared to random baselines, enabling efficient edge attacks on large graphs. Due to time and computational resource constraints, we were unable to include additional large-scale datasets. We will include a more detailed discussion on scalability in our revised version.
>
> > (2) The framework requires accurate estimation of the signal autocorrelation matrix, which may be challenging in real-world scenarios with limited or noisy data.
>
> We thank the reviewer for raising the concern about estimating the signal autocorrelation matrix. When graph signals are fully accessible, straightforward computation on the empirical autocorrelation is sufficient and reliable. In some challenging settings, such as when some node features are unobservable or corrupted by noise, more advanced statistical estimation techniques can be employed to obtain robust and accurate estimates.  For example, the graphical lasso estimator can be used to denoise the autocorrelation matrix, leveraging the assumption that the correlation structure among signals exhibits a certain level of sparsity. **We have already included a discussion of these alternative estimation strategies in Appendix C5** and, due to space limit here, kindly refer the reviewer there for a detailed discussion about alternative estimation methods.
>
> > (3) Evaluation is limited to homophily graphs
>
> We acknowledge that some other methodologies, such as DICE, are explicitly based on the homophily assumption.  In Section 4, we compare, under a homophilic setting (as captured by the cSBM), that our method exhibits similar behavior to DICE, even without access to ground-truth labels. However, we would like to emphasize that **our framework does not rely on the homophily assumption**. That setting is merely used as an easy-to-compare illustrative example. All of our theoretical results hold generally and are directly applicable to heterophilic or mixed-type graphs. The methodology, including the optimization formulation and the Prob-PGD algorithm, **remains valid regardless of the underlying homophily or heterophily structure**.
>
> To further clarify this point and address the concern, we include additional experiments on two benchmark heterophilic datasets: Chameleon and Squirrel. These datasets are known to exhibit low homophily scores and present challenges for traditional GNNs, and we adopt H2GCN, a GCNN model specifically designed for heterophilic graphs.
>
> H2GCN model details:
> Given a node feature matrix X and graph adjacency matrix A, H2GCN constructs node representations as $[X; AX, A^2X],$
> which is obtained by concatenating node feature X, aggregated feature with 1-hop AX, and aggregated with 2-hop neighbor $A^2X$. Such node embeddings are then passed through a multi-layer perceptron (MLP) to perform classification.
>
>
> We pre-train 10 models and report the mean+std of the prediction accuracy and embedding perturbation under different attack methods (1k edge perturbations for both datasets):
>
> (a) Chameleon dataset:
> - Nodes: 2277 (train:1092, val:729, test:456)
> - Edges: 36101
> - Feature dimension: 2325
> - Classes: 5
> - Homophily score: 0.23 (low, heterophilic)
>
> | Method      | Accuracy (%)           | Embedding Perturbation $\|\cdot\|_F$ |
> |-------------|------------------------|--------------------------------------|
> | Unperturbed (Test) | 55.92 ± 1.47     | -                                    |
> | Random      | 54.23 ± 1.16           | 136.65 ± 4.27                        |
> | Wst-PGD     | 54.87 ± 1.51           | 3099.12 ± 0                          |
> | Prob-PGD    | **52.96 ± 1.38**       | **6392.17 ± 0**                      |
>
> (b) Squirrel dataset:
> - Nodes: 5201 (train:2496, val:1664, test:1041)
> - Edges: 217073
> - Feature dimension: 2089
> - Classes: 5
> - Homophily score: 0.22 (low, heterophilic)
> | Method             | Accuracy (%)           | Embedding Perturbation $\|\cdot\|_F$ |
> |--------------------|------------------------|--------------------------------------|
> | Unperturbed        | 50.38 ± 0.39           | -                                    |
> | Random             | 47.91 ± 0.47           | 578.92 ± 3.86                        |
> | Wst-PGD            | 48.48 ± 0.73           | 1127.21 ± 0.00                       |
> | Prob-PGD           | **48.28 ± 0.73**       | **3908.26 ± 0.00**                   |
>
> In both heterophilic graphs, we observe that Prob-PGD consistently induces the largest embedding perturbation compared against the baselines. On the squirrel dataset, despite producing the largest embedding change, Prob-PGD does not result in the largest drop in prediction accuracy. We comment that this is not unexpected given that Prob-PGD is a **task-agnostic attack** that does not optimize for misclassification directly. Instead, it targets maximal disruption in node embeddings. As such, the relationship between embedding perturbation and accuracy degradation is not always linear or straightforward. For a more detailed discussion on this phenomenon, we refer the reviewer to our response to Reviewer f7yR.
>
> >(4)  adversarial defense against structural attacks?
>
> We thank the reviewer for the suggestion. In response, we evaluate and compare different attack methods on ElasticGNN [4], a robust GNN model that incorporates a locally adaptive smoothness regularization to enhance resistance against adversarial perturbations. We report results averaged over 10 independent runs:
>
> | Method       | Accuracy (%)       |
> |--------------|--------------------|
> | Test         | 81.57 ± 0.26       |
> | Random       | **80.20 ± 0.77**       |
> | Wst-PGD      | 80.79 ± 0.27       |
> | Prob-PGD     | 81.63 ± 0.46       |
>
>
> As shown in the results, the random baseline leads to the lowest post-attack accuracy, while our proposed method, Prob-PGD, results in the smallest performance drop. This indicates that Prob-PGD is currently less effective at degrading ElasticGNN's performance compared to other attack strategies.
>
> One possible explanation is that ElasticGNN is inherently robust to the class of perturbations introduced by Prob-PGD, which on homophilous graphs tends to behave like DICE and blur the community structure by connecting nodes from different classes. ElasticGNN, however, is particularly designed to tolerate such adversarial perturbations (see Section 4.3 in [4])
>
> This alignment between the behavior of ElasticGNN and Prob-PGD perturbations suggests that our attack stability framework may also **provide insight into why certain defense strategies are effective**. We plan to include more empirical studies to further explore and explain these connections.
>
> We emphasize that our current theoretical framework is primarily designed to analyze the stability of standard GNNs, rather than robustified architectures like ElasticGNN. While evaluating such models is valuable, extending our analysis to cover robust architectures would require separate, model-specific analysis. We consider this a compelling direction for future work.
>
> We would like to thank the reviewer again for their comments. We remain at their disposal during the discussion period and, in case the responses above addressed most concerns from the original review, would appreciate if the reviewer could kindly consider increasing the score/confidence in light of this.

---

> > ### Comment · Reviewer_SjSK · 2025-08-06
> >
> > Thank you for the thoughtful rebuttal. I appreciate the authors’ efforts to address the concerns raised in the initial review, particularly regarding autocorrelation estimation and the evaluation of heterophilic graphs.
> > However, I remain unconvinced by the scalability claims of the proposed method. The results on small-scale datasets like Cora with thousands of nodes are insufficient to demonstrate the practical viability of the approach, as data of such scale does not reflect the computational or structural challenges encountered in real-world graph applications nowadays. Moreover, the additional experiment on ogbn-arxiv, which compares the greedy version of the method only against a random baseline, does not provide compelling evidence of the scalability of the proposed method.

---

> > > ### Author Response · Authors · 2025-08-08
> > >
> > > We took the reviewer’s comment on scalability very seriously and apologise that our previous response was not entirely satisfactory. In addition to the preliminary results above, we have now had time to provide a more detailed response with more comprehensive results below.
> > >
> > > First of all, we would like to clarify that the motivation of including results on datasets such as Cora/MUTAG is that they serve as standard node-level or graph-level benchmarks; although they are relatively small-scale, graphs of similar size (e.g., hundreds or a few thousands nodes) can still be commonly found in graph-level tasks (e.g., most graphs in TUDataset are small for example) and many real-world applications such as biology, neuroscience, and finance (e.g., several hundreds of brain regions or stocks).
> > >
> > > Nevertheless, we agree with the reviewer that it would be important to test the scalability of our Prob-PGD attack in large-scale graphs. We already provided general directions of how to scale up our algorithm above, and for the experiments below we adopt an approximation of the algorithm by restricting the attack to deleting existing edges and greedily searching for edges to delete, where in each iteration of the greedy search we randomly sample candidate edges to delete (note that such a strategy is not uncommon in the graph learning literature; for example, both GraphSAGE and DropEdge employed similar strategies, albeit for different motivations). For ease of reference, we continue to refer to the adapted algorithms as Prob-PGD and Wst-PGD, though we note that both methods reported now are approximate versions designed for scalability. We conduct adversarial edge-deleting attacks by removing 200 existing edges, and **compare Prob-PGD against random baseline and Wst-PGD (using the same approximation technique for a fair comparison)**.
> > >
> > > We test on four datasets that are considered large-scale in the literature (according to [1] for example): ogbn-arxiv, ogbn-mag, ogbn-products, and pokec. (we skipped ogbn-proteins in [1] due to lack of node features). Additionally, ogbn-mag is an example of a heterogeneous graph (with different node types), while pokec is a graph with heterophily in node labels. Here, we report for each dataset its average embedding perturbation, computed as $1/d \sum_{i=1}^d \\|E_g x_i\\|_2$, where $x_i$ denotes the node feature and $d$ is the feature dimension.
> > >
> > > **(a) obgn-arxiv  (number of vertices=169,343, number of edges =1,166,243)**
> > >
> > > Embedding perturbation:
> > >
> > > Random: 4.0979
> > >
> > > Wst-PGD: 14.9566
> > >
> > > **Prob-PGD (Ours): 25.2936**
> > >
> > > - Peak memory usage (Prob-PGD): 390.58 MB
> > > - Running time (Prob-PGD): 537.74 seconds
> > >
> > > **(b) obgn-mag (number of vertices=1,939,743, number of edges=21,111,007)**
> > >
> > > Embedding perturbation:
> > >
> > > Random: 2.5874
> > >
> > > Wst-PGD:  8.5337
> > >
> > > **Prob-PGD (Ours): 15.7489**
> > >
> > > - Peak memory usage (Prob-PGD): 7362.41 MB
> > >
> > > - Running time (Prob-PGD): 1603.04 seconds
> > >
> > > **(c) obgn-products (number of vertices = 2,449,029, number of edges = 61,859,140)**
> > >
> > > Embedding perturbation:
> > >
> > > Random: 17.6481
> > >
> > > Wst-PGD: 24.3905
> > >
> > > **Prob-PGD (Ours): 193.8391**
> > >
> > > - Peak memory usage (Prob-PGD): 5833.25 MB
> > >
> > > - Running time (Prob-PGD): 2798.54 seconds
> > >
> > > **(d) pokec (number of vertices = 1,632,803, number of edges = 30,622,564)**
> > >
> > > Embedding perturbation:
> > >
> > > Random: 292.3226
> > >
> > > Wst-PGD:  303.3037
> > >
> > > **Prob-PGD (Ours): 334.1984**
> > >
> > > - Peak memory usage(Prob-PGD): 18040.25 MB
> > >
> > > - Running time(Prob-PGD): 1172.07 seconds
> > >
> > > From the results above, we can see that **Prob-PGD consistently outperforms both the random baseline and Wst-PGD in terms of embedding perturbation**. We hope these experiments demonstrate that our algorithm (with a reasonable approximation) can indeed scale up to graphs with millions of nodes and tens of millions of edges.
> > >
> > > We thank the reviewer again for their engagement and remain at their disposal for any further comments or suggestions they may have.
> > >
> > > [1] Classic GNNs are Strong Baselines: Reassessing GNNs for Node Classification, NeurIPS 2024.

---

> > > > ### Author Response · Authors · 2025-08-09
> > > > **Thank you for your engagement**
> > > >
> > > > Dear reviewer,
> > > >
> > > > We would like to express our sincere appreciation for your engagement in the review and rebuttal discussion, which we believe has helped improve the quality of the paper. We will incorporate the corresponding modifications in the final version accordingly.
> > > >
> > > > Best regards,
> > > >
> > > > Authors

---

### Decision · Program_Chairs · 2025-09-17

**Decision:**

Accept (poster)

**Comment:**

This work introduces a distribution-aware framework for analyzing the stability of graph convolutional neural networks to small graph topology perturbations with a probabilistic flavor. All the referees were mostly positive about this submission, and the rebuttal helped to improve some specific points of the paper. I recommend to the authors to carefuly implement every point raised by the referees in the camera ready version. I recommend acceptance to NeurIPS 2025.